# Room-temperature low-threshold avalanche effect in stepwise van-der-Waals homojunction photodiodes

Hailu Wang [1,2,7], Hui Xia [1,2,7 ✉], Yaqian Liu[1], Yue Chen[1,2], Runzhang Xie [1,2], Zhen Wang[1,2], Peng Wang [1,2], Jinshui Miao [1,2], Fang Wang [1,2], Tianxin Li [1,2], Lan Fu [3], Piotr Martyniuk[4], Jianbin Xu [5], Weida Hu [1,2 ✉] & Wei Lu [1,2,6 ✉]

Avalanche or carrier-multiplication effect, based on impact ionization processes in semiconductors, has a great potential for enhancing the performance of photodetector and solar cells. However, in practical applications, it suffers from high threshold energy, reducing the advantages of carrier multiplication. Here, we report on a low-threshold avalanche effect in a stepwise $WSe_2$ structure, in which the combination of weak electron-phonon scattering and high electric fields leads to a low-loss carrier acceleration and multiplication. Owing to this effect, the room-temperature threshold energy approaches the fundamental limit, $E_{thre} \approx E_g$, where $E_g$ is the bandgap of the semiconductor. Our findings offer an alternative perspective on the design and fabrication of future avalanche and hot-carrier photovoltaic devices.

Impact ionization is a process in which an electron or hole gains energy, e.g., by absorbing energetic photons or accelerating in a high-field region, to knock a secondary electron-hole pair out of the crystal lattice[1]. With the minimum threshold energy approaching the bandgap of the material ($E_{thre} \approx E_g$)[1-3], such an effect promises a performance boost in optoelectronic devices. Typically, the photovoltaic efficiency could overcome the Shockley–Queisser rule from 34% to 46%[4,5] and the detector's sensitivity reaches the photons' level at a low bias voltage[6]. However, in practical applications, it is hard to realize a threshold energy close to its minimum limit, leading to a low energy conversion efficiency during the carrier-multiplication process. For example, to activate the consecutive impact ionization in solar cells and avalanche photodetectors, the photon and electric-field energy must be 4 and 22 times higher than the bandgap energy, respectively[2,7].

Basically, there is an intense electron–phonon (e–p) interaction in traditional bulk materials[8]. It results in a huge waste of energy during the charge-carrier acceleration process, which thus delays the impact ionization process. Taking the InGaAs avalanche photodiode as an example, the room-temperature electron mean free path is ~140 nm (for InGaAs bulk material)[9], while the multiplication area is usually 1 μm in width[10]. It indicates that the electrons would take 7 times more chances of scattering during their acceleration process. A large amount of energy will then transfer to the lattice and dissipate in the form of phonon emissions.

In this work, we report on the room-temperature low-threshold avalanche effect in a $WSe_2$ homojunction. The avalanche threshold voltage is significantly reduced to ~1.6 V, which is at least 26 times lower than that of the traditional avalanche diode (e.g., InGaAs, the threshold voltage is up to 42 V)[10]. Furthermore, such device architecture shows a low background dark current (10–100 fA), and a high sensitivity (capable of detecting signals down to 24 fW, equivalent to $7.7 \times 10^4$ photons). All these characteristics indicate that an avalanche photodetector can be operated similarly to a conventional photodiode, enabling its utilization in a wide range of application scenarios.

[1]State Key Laboratory of Infrared Physics, Shanghai Institute of Technical Physics, Chinese Academy of Sciences, Shanghai 200083, China. [2]University of Chinese Academy of Sciences, Beijing 100049, China. [3]Department of Electronic Materials Engineering, Research School of Physics and Engineering, The Australian National University, Canberra, ACT 2601, Australia. [4]Institute of Applied Physics, Military University of Technology, 2 Kaliskiego St., 00-908 Warsaw, Poland. [5]Department of Electronic Engineering and Materials Science and Technology Research Center, The Chinese University of Hong Kong, Hong Kong SAR, China. [6]School of Physical Science and Technology, ShanghaiTech University, Shanghai 201210, China. [7]These authors contributed equally: Hailu Wang, Hui Xia. ✉e-mail: huix@mail.sitp.ac.cn; wdhu@mail.sitp.ac.cn; luwei@mail.sitp.ac.cn

## Results

### Principle of the low-threshold avalanche

Two-dimensional transition metal dichalcogenide (TMD) materials are selected as the gain medium for charge-carrier avalanche. Lately, Barati et al. stated that the e−p coupling is intrinsically low in $WSe_2/MoSe_2$ heterostructure, which leads to a slow cool-down of hot carriers and thus triggers the interlayer charge-carrier multiplication[11]. Kim et al. reached a similar conclusion by determining the carrier-multiplication efficiency as 99% in $MoTe_2$ and $WSe_2$ films[2]. One explanation for such a distinct property is described in Fig. 1a, b. The phonons in TMD materials are classified into two groups, out-of-plane, and in-plane modes[12]. Generally, (i) the out-of-plane modes (e.g., $A_{1g}$ mode) are more active in the e−p interactions (as compared with the in-plane modes, e.g., $E_{1g}$ and $E_{1u}$); (ii) the contribution of out-of-plane mode to e−p scattering could be significantly weakened by thinning the material[13–16]. Thus, as verified in Raman and inelastic-electron-tunneling spectroscopy experiments, there is the possibility to minimize the e−p scattering, e.g., by thinning the TMD material to the monolayer limit[17,18].

Additionally, we shape the TMD materials into a stepwise geometry during the standard mechanical exfoliation process. There is also an alternative routine to carve the shape of TMD materials, e.g., by selective area dry-etching process, more details can be found in Supplementary Figs. 1, 2. Naturally, the morphology transition gives rise to an atomically abrupt homojunction, in which the few-/multi-layer segment serves as a wide-/narrow-gap semiconductor, respectively[19]. More notably, such device architecture accumulates more electric-field energy at the depletion region and thus triggers the charge-carrier avalanche process earlier. According to the numerical simulation (see more details in the "Methods" section), the peak electric field of the vdW junction is about 4 times higher than that of the traditional counterpart (Fig. 1c, d). Consequently, the external bias voltage required for avalanche would be almost 16 times lower, e.g., scaling down from −24.5 to −1.6 V (Fig. 1e), determined by the relation: $E \propto (V_{ex})^{1/2}$, where $E$ and $V_{ex}$ represent the electric field and external bias voltage, respectively[20].

Note that in bulk material systems, the stepwise or mesa-island morphology is undesirable due to the resulting poor photoelectric performance. First, it produces a mass of defects/damages at the multiple boundary surfaces (mostly coming from the wet or dry-etching process) that could lead to a large leakage current and a short minority-carrier lifetime[21]. Second, the geometric singular point (especially those right-angled corners) would accumulate electric power and result in a destructive breakdown[22]. For the layered material, however, the vertically stacked layers are bonded by weak van-der-Waals force. It spares the material from most surface and interface issues[23]. Therefore, we can shape the device into any wanted morphology and make use of it to enhance the performance.

### Device structure and its fabrication process

As stated above, the stepwise vdW junction is featured by the weak e−p interaction and an enhanced electric field, both of which should benefit the charge-carrier avalanche process. To test this idea, we developed the stepped $WSe_2$ avalanche devices. As shown in the inset of Fig. 2a, the stepwise $n^-$-$WSe_2$ flake was mechanically exfoliated onto a $SiO_2/Si$ substrate in advance, and the electrical contacts were ensured by depositing Pt/Au electrodes on both sides (see more details in "Methods" section). Figure 2b shows the high-angle annular dark-field scanning transmission electron microscope (HAADF-TEM) and energy-dispersive X-ray (EDX) images of the device. The morphology transition between few- and multi-layer $WSe_2$ is atomically abrupt and the thickness of them is determined as 4 layers (L) and 39 L, respectively. We also fabricated more than 25 devices based on different thickness combinations, in which the few-layer thickness spans from 3 to 13 L and the multi-layer thickness spans from 13 to 75 L. A statistical analysis of the geometrical configuration and photoelectric performances of

those devices can be found in Supplementary Figs. 3–10 and Supplementary Table 1.

Figure 2c shows the Raman spectra of 4 and 39 L $WSe_2$ measured with a 532 nm laser line. The feature peaked at 249.3 $cm^{-1}$ is the first-order Raman signal, which arises from the $A_{1g}$ phonon mode[24]. Additional features peaked at 138.8, 257.5, 309.5, 360.6, 373.1, and 395.7 $cm^{-1}$ corresponding to the second-order Raman signals, that are associated with combination and overtones of phonons[25]. Obviously, the second-order Raman signals decline as the thickness of $WSe_2$ scales from 39 to 4 L, e.g., the $2LA(M)$ (peaked at 257.5 $cm^{-1}$) intensity declines from 67% to 24% of the $A_{1g}$ intensity. It is consistent with the Raman results from previous reports[24,25]. Such character indicates that there are fewer phonons active in $WSe_2$ as its thickness shrinks to the monolayer limit. This might be a clue to understanding the intrinsic weak e−p interaction property of TMD materials.

### Photoelectric properties of stepwise WSe₂ diodes

Figure 2a shows the dark- and photo-excited I−V curves of the A# device. The measurements were performed at room temperature and the external bias voltage was applied to the multi-layer $WSe_2$. The dark I−V curve can be divided into two distinct regions, including rectifying and breakdown areas. In the rectifying region, −1.44 V < $V_{ex}$ < 3 V, the cut-off current is down to 10 fA and the rectification ratio is up to $10^3$. It is easy to understand the rectifying property since the band offset between few- and multi-layer $WSe_2$ leads to an internal electric field. A detailed discussion of such characteristics can be found in Supplementary Fig. 11. In the breakdown region, −3 V ≤ $V_{ex}$ ≤ −1.44 V, the current boosts to a high level. We compare the I−V curves of the $WSe_2$ device with those of commercial InGaAs avalanche diodes. As shown in Supplementary Fig. 12, both kinds of devices experience a ~$10^4$ times increase of current after breakdown. And, more importantly, the current of the stepwise $WSe_2$ device climbs at the same rate as that of the InGaAs avalanche device, $dV/d(\lg I) \approx 400$ mV/dec. What interests us is that there is a high photogain after breakdown (Fig. 2d). It allows the device to detect light signals down to the femtowatt level. A more detailed discussion of the photosensitivity of the stepwise $WSe_2$ diode is performed in the subsequent sections.

### Threshold limit of WSe₂ diodes at low temperature

To explore the nature of the current breakdown, we further performed low-temperature experiments on different stepwise $WSe_2$ diodes (A# and B# devices). As depicted in Figs. 2e and 3a, b, the devices exhibit a positive temperature coefficient, in which the reverse breakdown voltage ($V_{br}$) decreases as the temperature falls (consistent with the phenomenon observed in the conventional InGaAs avalanche diodes, Supplementary Fig. 13). For devices A# and B#, $V_{br}$ transits from ~−1.44 V at 300 K to ~−0.70 V at 240 K and ~−1.61 V at 292 K to ~−0.70 V at 170 K, respectively. It helps to confirm that the current breakdown indeed comes from the avalanche process. The underlying physics is described as follows. As temperature falls, electrons are gradually spared from phonons scatterings (by freezing the lattice vibration), and thus easily gain an excess energy of $E_g$ to initiate the impact ionization process. For this reason, the external electric voltage/power required for the avalanche is decreased continuously[26]. By contrast, the forward onset voltage ($V_{on}$) exhibits a negative temperature coefficient (Fig. 3b), that is associated with the broadened bandgap at low temperatures.

According to the equation, $I_{forward} \propto e^{\frac{(qV_{ex}-E_g(T))}{kT}}$[27], the forward current would not exponentially increase unless the external bias voltage is comparable to $E_g(T)/q$. Thus, the onset voltage would be proportional to the bandgap of the semiconductor, and the threshold voltage would increase accordingly at low temperatures. Herein,

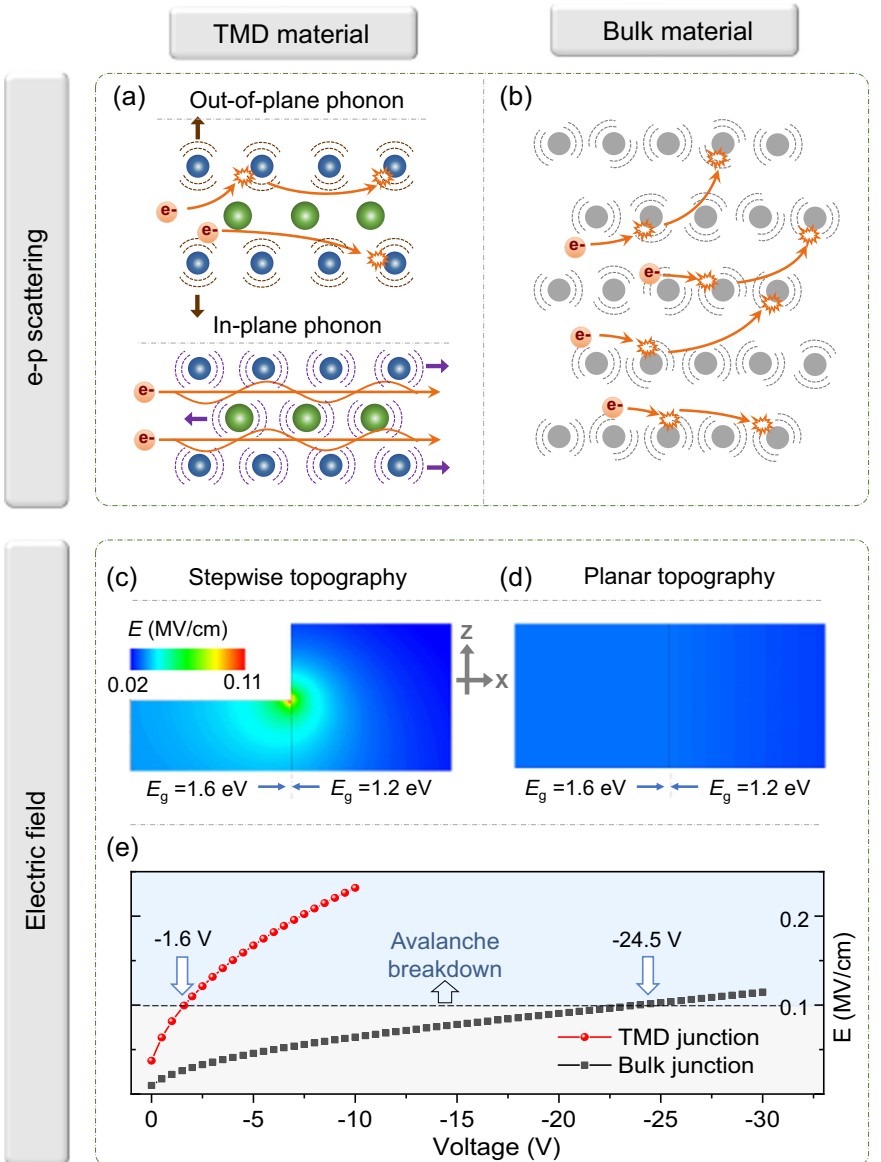

**Fig. 1 | Transition metal dichalcogenide (TMD) and bulk materials for charge-carrier avalanche.** Sparse and intense electron–phonon scatterings in (**a**) TMD and (**b**) bulk materials, respectively. In TMD materials, the phonons are classified into out-of-plane and in-plane modes, where the former/latter contributes most/little to the electron scatterings. Simulated electric-field distribution in the $x$–$z$ cross-section of (**c**) TMD and (**d**) bulk junctions, respectively. The TMD junction is featured by a stepwise topography, with the bandgap of few- and multi-layer segments set as 1.6 and 1.2 eV, respectively. The bulk junction is characterized by a planar topography, with the bandgap of left and right half segments set as 1.6 and 1.2 eV, respectively. The vertical lines in the color plots and blue arrows clearly show the boundary of the two bandgap counterparts. The color bar is unified for the two color plots. **e** Dependence of peak electric field on the reverse-biased voltage for TMD (red scatter line) and bulk junction (black scatter line). The dashed line indicates the electric-field intensity that usually requires for a charge-carrier avalanche.

$I_{forward}$ represents the forward current, $q$ is the elementary charge, $k$ is the Boltzmann constant and $T$ is the temperature.

Figure 3c shows the dependence of $V_{br}$ on the operating temperature. Obviously, the temperature coefficient ($\Delta V_{br}/\Delta T$) decreases significantly, especially when the temperature goes below 250 K. It directly leads to a saturation tendency of the breakdown voltage. Such character indicates that there is a final limit of $V_{br}$ as temperature goes down. More results measured on different devices are shown in Supplementary Fig. 14, which verifies that this is a common behavior. By fitting the curve with a double-exponential equation: $V_{br} = -A \times e^{\frac{T-T_0}{\alpha}} - B \times e^{\frac{T}{\beta}}$, the final limit value, $-B$, is derived as $-0.57$ V ($0.35E_g/q$). Herein, $T$ represents the operating temperature, A, α, β, and $T_0$ are constant and fitted as 0.042 V, 28 K, 850 K, and 209 K. To understand the underlying

physics, we further performed numerical simulation and scanning kelvin probe microscopy experiments on the $WSe_2$ diode. Generally, there are two voltage components contributing to the electron acceleration, external bias voltage and internal built-in potential. The external bias voltage is up to $100E_g/q$ in Si and GaN avalanche devices, while the built-in potential is at a low level, $-0.7E_g/q$. This makes people easily neglect the contribution of the latter. In a low-threshold avalanche device, however, the breakdown voltage is lowered to $-E_g/q$. In this regard, the contribution of built-in potential to the avalanche performance should not be neglected. As shown in Supplementary Fig. 15 and Supplementary Table 2, the built-in potential arises from two aspects: the band-offset and the Fermi-level drop (induced by different doping polarities and concentrations in the counterpart segments). According to the simulation, the band offset solely leads to an internal potential of 0.41 V,

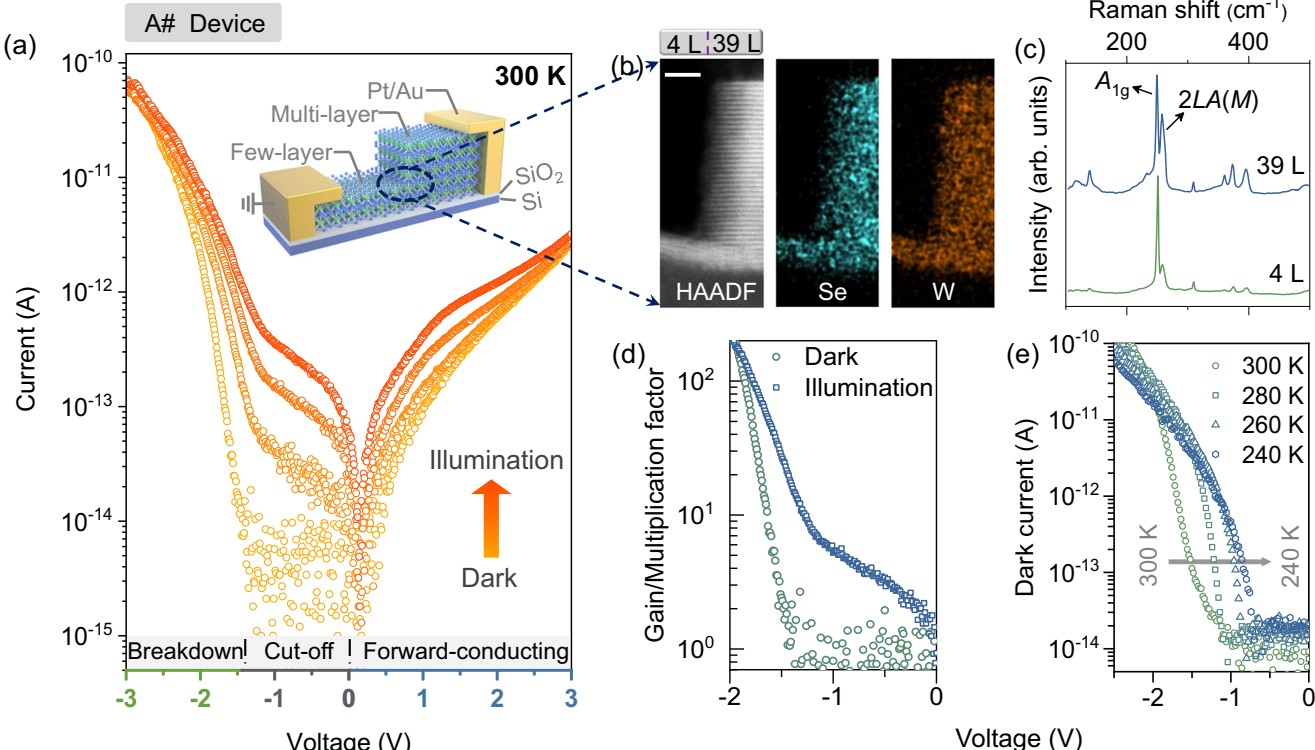

**Fig. 2 | Stepwise TMD diode and its optoelectronic property. a** Dark and photoexcited *I–V* curves of A# device measured at room temperature. The photoexcited *I–V* curves were obtained under a 520 nm laser illumination with an increasing intensity from 2.52 nW to 9.97 nW and 25.78 nW. The diode is at forward-conducting, cut-off and reversed breakdown state as it is operated at $0 < V_{ex}$, $-1.6\,V < V_{ex} < 0\,V$, and $V_{ex} < -1.6\,V$, respectively. Inset: schematic showing the device structure. **b** High-angle annular dark-field transmission electron microscope (HAADF-TEM) and energy-dispersive X-ray spectroscopy (EDX) images of A#

device. The scale bar is 5 nm. **c** Raman spectra of 4 layers (L) and 39 L WSe$_2$ measured with a 532 nm laser line. The spectra are normalized to the $A_{1g}$ peak and vertically offset for clarity. **d** Multiplication factor/gain derived in dark and under illumination, according to the equation $G = I_d/I_{bg}$ and $G = (I_{ph} - I_d)/I_{bg}$, where $I_{ph}$ represents the photocurrent, $I_d$ is the dark current and $I_{bg}$ denotes the dark-current/photocurrent when $G = 1^6$. **e** Reverse-biased *I–V* curves of the A# device as the temperature drops from 300 to 240 K.

while the latter one rises it to 1.18 V and even higher. If we assume that there is no energy loss during the electron acceleration process, the minimum voltage required for the avalanche would be lowered to $\frac{E_g = 1.6\,eV}{q} - 1.2\,V = 0.4\,V$ (Supplementary Table 2). The scanning kelvin probe microscopy experiments also support such judgment. As shown in Fig. 3d, e, the built-in potential drop between few- and multi-layer WSe$_2$ is proved to be up to 1.1 V. This feature helps explain why the breakdown voltage declines further as the temperature goes down.

### Locating the avalanche multiplication area

Recently, there have been reports stating that the vdW avalanche easily takes place at Schottky contact[28–30]. To clarify this kind of issue, we performed scanning photocurrent mapping experiments (SPCM) on the stepwise WSe$_2$ diode. Considering that the work function of the Pt electrode ($\Phi_{Pt} = 5.65\,eV$) is close to that of WSe$_2$ ($\Phi_{WSe_2} = 5.43\,eV$), the effect of the Schottky barrier on the photocarrier harvest could be neglected. The SPCM results help to verify such judgment. As depicted in Fig. 4a, no photoresponse is observed at the metal–WSe$_2$ interface, indicating a little Schottky barrier there. We also show the SPCM pattern of the Schottky vdW device for comparison. The reference device is fabricated on a uniform WSe$_2$ flake, with Cr/Au electrodes deposited on both sides. Considering that the work function of Cr ($\Phi_{Cr} = 4.5\,eV$) is much lower than that of WSe$_2$, there are large Schottky barriers and thus photoresponse at the metal–WSe$_2$ interface, shown in Fig. 4b. Those results demonstrate that the avalanche observed here originates from the WSe$_2$ homojunction.

### Figure-of-merit of stepwise WSe$_2$ avalanche diodes

Figure 5a and Supplementary Table 3 summarize the room-temperature breakdown voltage of bulk, van-der-Waals material, and our stepwise WSe$_2$ structure. Obviously, the threshold voltage is up to 150 V in bulk material. It poses severe requirements on material, operation, and signal processing. For instance, there are few choices of materials to fabricate such kinds of devices, since they must be clean enough (background doping concentration in $\sim10^{15}\,cm^{-3}$) to bear the high electric field (0.1–1 MV/cm) without destructive breakdown[7,31]. Meanwhile, the driving and signal-processing circuit should be specially designed due to the ultra-high driving voltage (up to 150 V). These requirements dramatically increase the cost and thus limit the application scenarios. Recently, inspired by the intriguing feature of vdW materials, avalanche devices based on uniform InSe[32], MoS$_2$[33,34], WSe$_2$[35,36] and BP[37] flake, WSe$_2$/MoS$_2$[38], Graphite/InSe[28], and BP/InSe[6] heterostructures have been investigated. Those efforts reduce the avalanche voltage to ~10 V at 300 K, which stands for a big step forward. However, the breakdown voltage is still out of the range of traditional digital circuits (±5 V voltage range). In this work, we show that the stepwise WSe$_2$ architecture could further reduce the threshold voltage to 1.6 V (Fig. 2a and Supplementary Table 1). It reaches the classic limit of avalanching at room temperature since the external energy required for avalanching equals the energy gap of multiplication region, $V_{ex} \times q \approx E_g = 1.6\,eV$ (for bulk avalanche diode, the threshold energy must reach 22$E_g$ and more). This means that the thermodynamic loss is low during the charge carriers' acceleration process, which thus relaxes the restriction on the external electric power. As a return, the common digital circuits can be used to drive

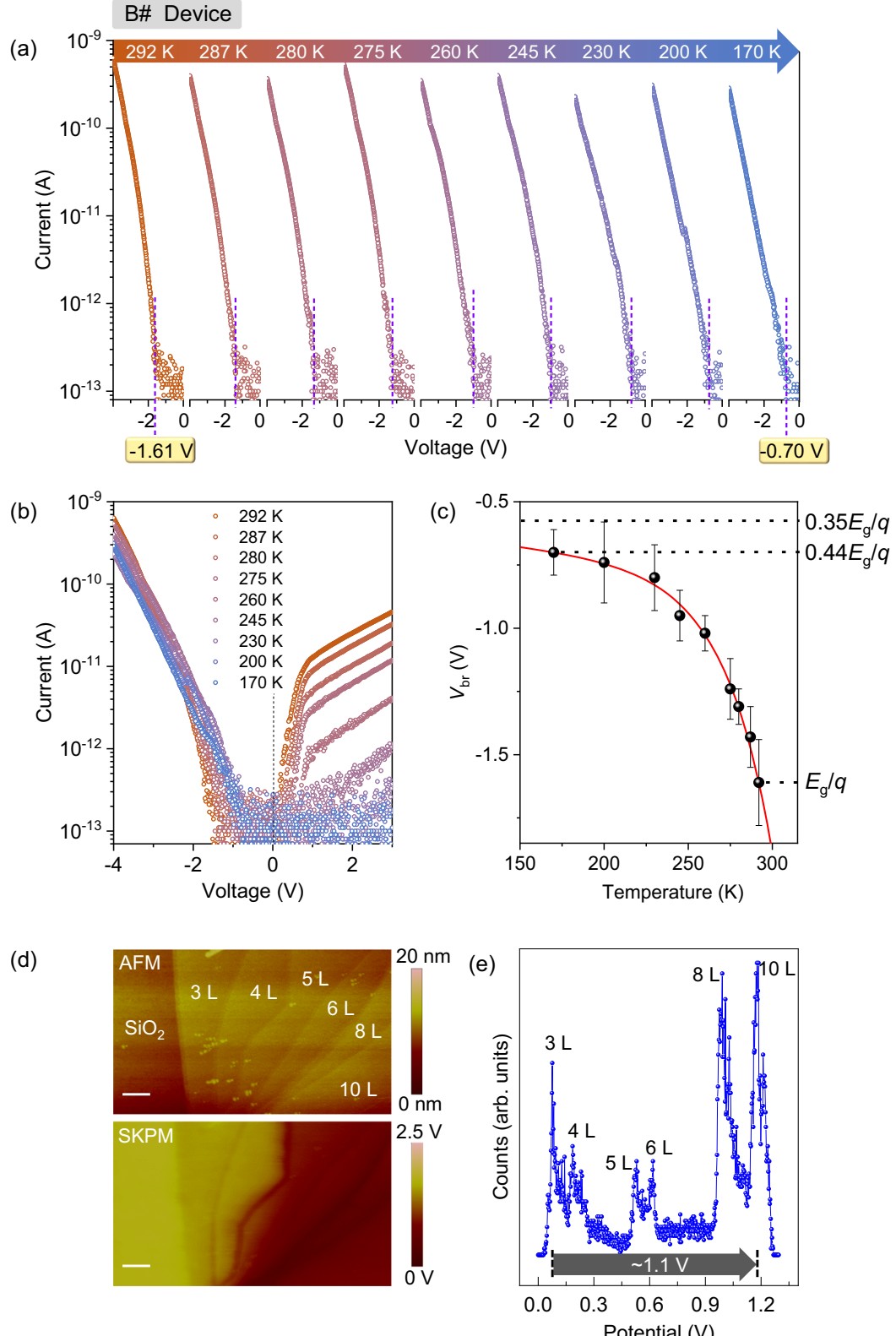

such avalanche diodes, which will greatly extend the application scenarios.

In a traditional avalanche diode, the dark current rapidly catches up with the photocurrent after avalanching. It leads to difficulty in resolving signals from background noise in Geiger mode[39]. Here we show the advantage of the WSe₂ diode in avalanching applications. As depicted in Supplementary Fig. 16, the photocurrent is ahead of the dark-current avalanche. It results in a high photogain, up to 470. At the

same time, the ratio of photocurrent to dark-current (signal-to-noise ratio) reaches a high level, up to 167. Those characters allow the device to conveniently work at Geiger mode and detect light signals down to the femtowatt level. Experimentally, the lowest illumination intensity that the device can respond to is ~24 fW. Considering that the laser wavelength is 637 nm, the lowest phonon number that the device can detect is ~7.7 × 10⁴. The actual imaging measurements of the "SITP" letter graphics were also performed at room temperature to further

**Fig. 3 | Threshold limit of avalanche.** Reverse-biased (**a**) and full-scale (**b**) I−V curves of B# device that was operated at decreasing temperatures. The dashed line in (**b**) indicates the zero-bias point that helps to distinguish the reverse bias region from the forward bias region. **c** Dependence of reverse breakdown voltage on the operating temperature. The scatter lines are the experimental data derived from (**a**). The error bar shows the error in determining the breakdown voltage by identifying the current breakdown point from the I−V curves. The solid line is the fitting curve according to a double-exponential equation: $V_{br} = -A \times e^{\frac{T-T_0}{\alpha}} - B \times e^{\frac{T}{\beta}}$, where $V_{br}$ represents the breakdown voltage, T represents the operating temperature, A and B are constant,

α, β, and $T_0$ are reference temperature. The upper dashed line indicates the final limit of avalanche voltage, that is fitted as −0.57 V ($0.35E_g/q$, $E_g$ is the bandgap of the few-layer WSe₂, q is the elementary charge). The middle dashed line represents the minimum avalanche voltage that was derived in the experiment, −0.70 V ($0.44E_g/q$). The bottom dashed line represents the avalanche voltage that was derived at room temperature, −1.61 V ($E_g/q$). **d** Atomic force microscope (AFM) and scanning kelvin probe microscopy (SKPM) images of a WSe₂ flake that consists of six different layer segments, 3, 4, 5, 6, 8, and 10 L. The scale bar in AFM and SKPM images is 1 μm. **e** Surface potential of the six different layer segments derived from (**d**).

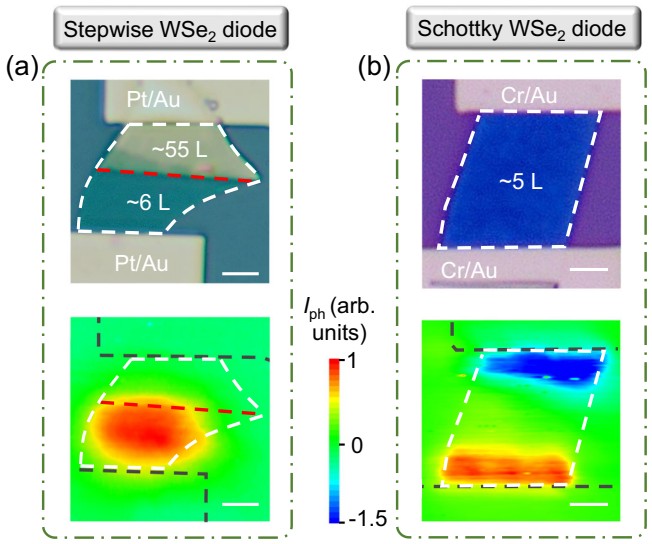

**Fig. 4 | Scanning photocurrent mapping (SPCM) experiments on WSe₂ diodes.** Optical microscope and corresponding SPCM images of (**a**) stepwise WSe₂ and (**b**) Schottky WSe₂ diodes. The scale bars in (**a**) and (**b**) are 4 and 2 μm, respectively.

validate the photoelectric performance of the WSe₂ photodiode, which exhibited fast response time and high stability. More details can be found in the Supplementary Figs. 17–20.

Figure 5b shows the avalanche process in the stepwise WSe₂ diode and explains why the photocurrent is ahead of the dark avalanche. The band structure displayed here is derived at −2 V by rigorous simulation. In dark conditions, the electrons are swept away from the high-field region (heterojunction interface). Thus, they have to accelerate over a long distance, ΔD, before getting an excess energy of $E_g$. Under illumination, however, the photogenerated carriers (mainly the minority holes generated at the few-layer segments, as demonstrated by the SPCM experiments) can accelerate through the high-field region. ΔD is then reduced to half that of the dark case, which favors the impact ionization process. Those characters lead to the offset between dark-current and photocurrent after the avalanche.

We further calculate the impact ionization rate of the stepwise WSe₂ device, by solving the equation, $1 - \frac{1}{M} = \int_0^W \alpha \left[ \exp \left( \int_0^x -\alpha dx \right) \right] dx$, where M is the multiplication factor, α is the impact ionization rate, and W is the width of the channel or multiplication region[27]. Considering that the avalanche process mainly arises from the hole impact ionization in stepwise WSe₂ diode, the rate calculated here is thus the hole impact ionization rate. Figure 5c summarizes the hole impact ionization rate of bulk material[40,41], uniform WSe₂[36], and our stepwise WSe₂ avalanche devices. One can clearly find that the bulk material requires an ultra-high uniform electric field, $2 \times 10^5 - 1 \times 10^6$ V/cm, to raise the impact ionization rate to a level of $10^4 - 10^5$ cm⁻¹. In the uniform WSe₂ materials, by contrast, the electric field required for an avalanche is lowered by ~10 times. And in our stepwise WSe₂ avalanche devices, it is further reduced by 20

times, to a low value (herein, the electric field is assumed uniform in the stepwise device for convenience of calculations).

In the devices present above, the WSe₂ material is exposed to the SiO₂/Si substrate, which might suffer from the scattering from the substrate[42]. To clarify this kind of issue, we fabricated additional WSe₂/hexagonal boron nitride (hBN) devices. As shown in Supplementary Fig. 21, an hBN film was first transferred onto the SiO₂/Si substrate, followed by a dry transfer of a stepwise WSe₂ layer. In this way, the WSe₂ material is isolated from the substrate. The electrode configuration is the same as that of bare WSe₂ devices. As summarized in Supplementary Figs. 22–25, the breakdown voltage of all 11 WSe₂/hBN devices falls in the range of −1.2 to −1.8 V, while that of bare WSe₂ devices shows a quite discrete distribution in a wide voltage range, −1.4 to −5.4 V. For a more detailed analysis, we divided the threshold voltage into three ranges: $|V_{br}| < 1.5$ V, $1.5$ V $\leq |V_{br}| < 2$ V, and $|V_{br}| \geq 2$ V. Among them, the proportions of WSe₂/hBN and pristine WSe₂ diodes in the three threshold voltage ranges are 36%, 64%, 0% and 20%, 40%, 40%, respectively. This further confirms that the overall performance of WSe₂ diodes is indeed improved when an hBN layer is used as the substrate, due to the reduced scattering processes.

## Discussion

Our stepwise layer junction approaches the limit of the avalanche effect, where the room-temperature threshold energy is approximately equal to the bandgap of the semiconductor ($E_{thre} \approx E_g$). Such distinct property arises from the weak e−p interactions as well as an enhanced electric field. For this reason, we can operate the van-der-Waals avalanche device just like a diode or transistor with the help of an ordinary digital circuit (the driving voltage range is ±5 V). By contrast, traditional avalanche devices cannot function without high-voltage accessories (the driving voltage is up to 150 V). Also, the stepwise layer junction shows a low dark current (10–100 fA) in the linear region, which is 4 orders of magnitude lower than that of traditional avalanche devices. At the same time, such a device can sense light signals with its intensity down to 24 fW, that is $7.7 \times 10^4$ photons in the count. Together, those characteristics indicate the great potential of van-der-Waals avalanche devices in future low-cost optical communication scenarios. The findings disclosed here are also valuable to photovoltaic applications. Because, in such a framework, the thermodynamic loss is extremely low. The high-energy photons ($E_{ph} \geq 2E_g$) thus have more chances to create two electron-hole pairs (by activating the carrier-multiplication effect), which might substantially improve the photovoltaic efficiency.

## Methods

### Numerical simulation

The theoretical model is established with a commercial software package (Sentaurus-TCAD). It considers the stepwise geometry of the TMD device, where the two distinct segments are set as 5.6 (8 L) and 20.3 nm (29 L) in thickness. The electron affinity and bandgaps are 3.7 and 1.6 eV, 4.0 and 1.2 eV, for few- and multi-layer parts, respectively[43,44]. A uniform background doping concentration is assumed and set to be $1 \times 10^{15}$ cm⁻³. Following the experimental setup,

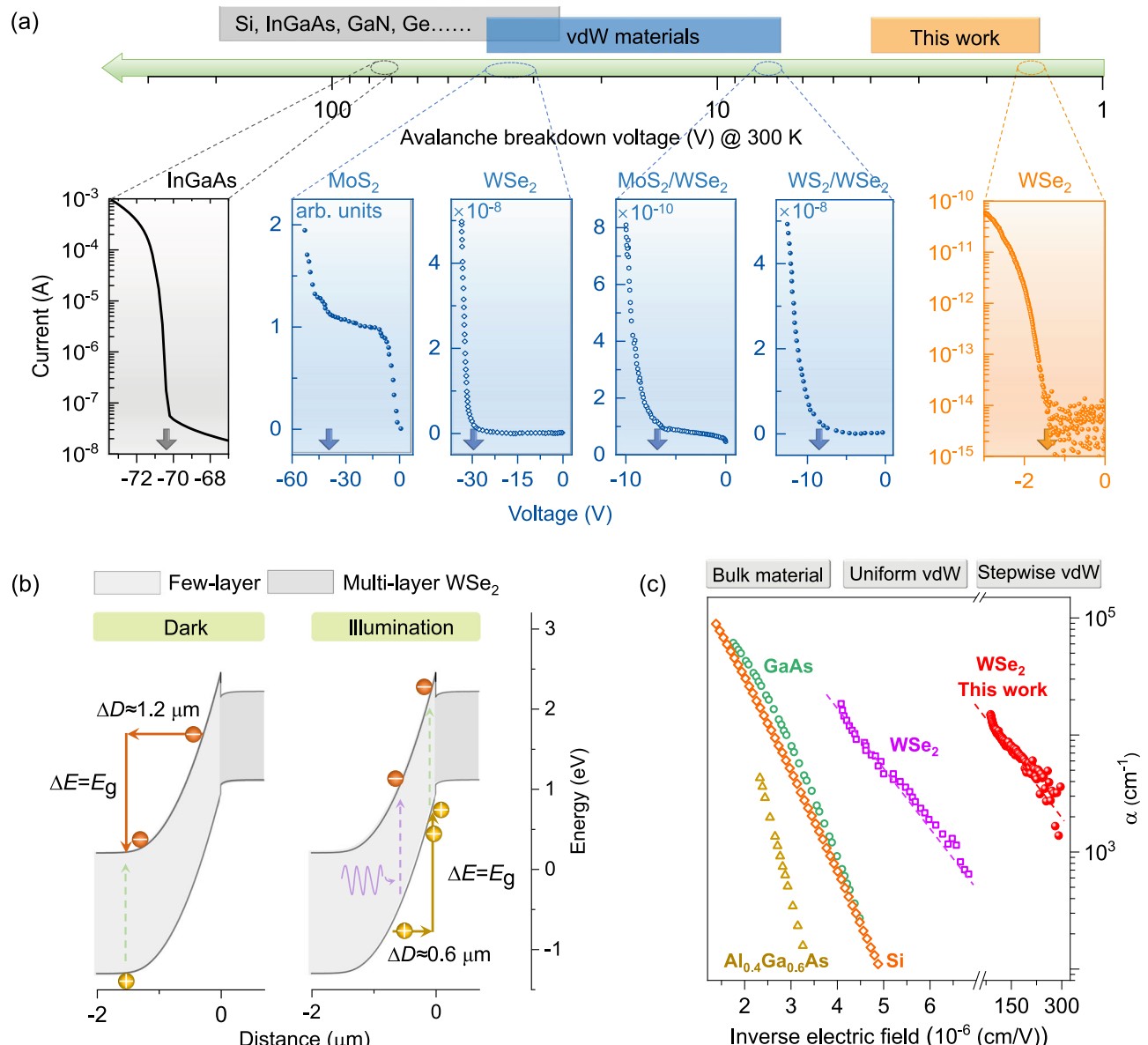

**Fig. 5 | Figure-of-merit of the stepwise TMD avalanche diode. a** Summary on the breakdown voltage of WSe₂ and traditional avalanche diodes. The arrows in the six *I–V* curves indicate the breakdown voltage. The middle four panels: reprinted (adapted) with permission from ref. 33. Copyright (2018) American Chemical Society and from ref. 35. Tsinghua University Press, 2022, reproduced with permission from SNCSC. Reprinted (adapted) with permission from ref. 38. Copyright (2022) American Chemical Society. **b** Schematic showing the distinct charge-carrier avalanching processes in the dark and under illumination for the TMD avalanche diode. $\Delta D$ denotes the acceleration distance that electrons and holes require to get an excess energy of $\Delta E$ ($E_g$). As a unilateral depletion junction, the charge-carrier acceleration process mainly takes place at the few-layer segment. **c** Summary on the hole impact ionization rate of bulk material (reprinted from ref. 40, copyright (1973), with permission from Elsevier. Reprinted from ref. 41, with the permission of AIP Publishing), uniform WSe₂ (reprinted (adapted) with permission from ref. 36, copyright (2022) American Chemical Society), and our stepwise WSe₂ avalanche devices.

the bias voltage is applied to the multi-layer section while the few-layer part is kept ground. Finally, the band-alignment and internal electric-field distribution are calculated by a coupled solution of the Poisson equation, and electron and hole continuity equations.

The simulation considers a uniform morphology, rather than a stepwise feature, for the bulk junction. It is reasonable since the stepwise or mesa-island morphology is usually taken as the performance killer in bulk avalanche devices. Typically, the high density of surface defects/damages at the side wall (mostly coming from the wet or dry-etching process) could lead to a large leakage current and even destructive breakdown[21,45]. Thus, the thickness of narrow and wide semiconductors is kept consistent at 5 nm. Apart from that, the

parameters are the same as those of the TMD model, such as the charge-carrier density, electron affinity, and bandgap.

## Material characterization and device fabrication

The WSe₂ flake was mechanically exfoliated from an $n^-$-doped WSe₂ crystal (purchased from Shanghai Onway Technology Co., Ltd) and transferred onto an Si substrate with a 300 nm thick SiO₂. Electron-beam lithography process was then carried out by FEI F50 with an NPGS system, to depict the electrode patterns. A metal contact of 20 nm Pt/70 nm Au was deposited by the dual-ion beam sputtering method, which is followed by the standard lift-off processes. The Raman spectra of few- and multi-layer WSe₂ were obtained by a Lab

RAM HR 800 with an exciting laser of 532 nm. Tecnai F20 TEM was used to reveal the morphology of the stepwise WSe$_2$ diode.

## Photoelectric characterization

The electrical and photoelectric measurements were performed by a probe station (Lake Shore TTPX) combined with a semiconductor parameter analyzer (Keysight B1500A). An external laser source (Thorlabs LP520-SF15) controlled by a diode current and temperature controller (Thorlabs ITC4200) are introduced to excite the devices. The incident light intensities were determined by the power meters (Thorlabs S1300VC and Newport 1936-C). In the temperature-dependent measurements, the sample temperature was held by a cooling system that consists of a constant liquid nitrogen flow, an electrical heater, and a temperature controller.

## Data availability

The Source data underlying the figures of this study are available with the paper and are accessible at https://doi.org/10.6084/m9.figshare.25578000. All raw data generated during the current study are available from the corresponding authors upon request. Source data are provided with this paper.

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

## Acknowledgements

This work was supported by the Strategic Priority Research Program of the Chinese Academy of Sciences (Grant Nos. XDB0580000, W.H.; XDB43010200, T.L.), National Natural Science Foundation of China (Grant Nos. 62327812, W.H.; 11991063, T.L.; 12174416, H.X.; 62304231, H.W.; 12393833, H.X.; U2241219, H.X.), Shanghai Science and Technology Committee (Grant No. 23WZ2500400, W.H.), International Partnership Program of Chinese Academy of Sciences (Grant No. 181331KYSB20200012, W.H.), China National Postdoctoral Program for Innovative Talent (Grant No. BX20230390, H.W.), and China Postdoctoral Science Foundation (Grant No. 2023M733622, H.W.).

## Author contributions

H. Xia, W. Hu and W. Lu supervised the project, proposed the idea, and designed the experiments. H. Wang and Y. Liu fabricated the device. H. Wang and H. Xia performed the (opto-) electrical experiments. H. Xia carried out the numerical simulations. H. Xia and H. Wang analyzed the data and prepared the manuscript. Y. Chen performed the AFM measurements. R. Xie, Z. Wang and P. Wang helped to analyze the photoelectrical data. J. Miao, F. Wang, T. Li, L. Fu, P. Martyniuk and J. Xu discussed and commented on the manuscript.

## Competing interests

The authors declare no competing interests.
