## [Peer Review File · Nature Communications]

Room-temperature low-threshold avalanche effect in stepwise van-der-Waals homojunction photodiodesEditorial Note: Parts of this Peer Review File have been redacted as indicated to remove third-party material where no permission to publish could be obtained.

REVIEWER COMMENTS

Reviewer #1 (Remarks to the Author):

In this study, the authors present work on using 2D vdW homojunctions for observing an avalanche breakdown under a low voltage bias that is close to that of the threshold limit (i.e. electrical energy that corresponds to $\sim E_g$). Homojunction is composed of lateral few-layer/multi-layer junction naturally formed from mechanical exfoliation due to the different thicknesses of the exfoliated flake. The homojunction device is fabricated by depositing each Pt/Au electrode on each region (i.e. few-layer and multi-layer region of the flake). The authors attribute a significant increase in the current at a reverse-bias voltage to avalanche breakdown effect and discuss the breakdown voltage in terms of the onset of such large increase in the current. Some qualitative arguments are shown for explaining the temperature-dependence and the shift of the VOC according to the incident light intensity. In many aspects, I believe that many of the claims made by the authors in this manuscript are overstatements often lacking physical bases to clearly demonstrate avalanche breakdown in their 'homojunction' device. Therefore, I recommend rejection of this manuscript to as highly respected journal as Nature Communications and seriously recommend authors to thoroughly re-interpret their results. In details, I find their manuscript should not be submitted to another journal without re-interpreting their results without their unsubstantiated explanations based on avalanche breakdown and considering the points further below:

1. Homojunction: the authors should not use the word 'homojunction' for the device architecture that they propose; homojunctions typically consist of the same semiconductors but different doping levels (e.g. p-n junctions in Si). If they understand that the few-layer and multi-layer WSe₂ have different bandgaps due to their different electronic structure. It is also disappointing that the authors have not specified their relevant electronic structures; indirect or direct bandgap, which prompts the discussion of whether their claim of having 'reduced el-ph coupling' can be physically justified in the context of avalanche breakdown (which is not strictly proven, either).
2. Avalanche carrier multiplication mechanism: I can not find qualitative or quantitative supporting evidence for claiming that the observed increase of current under the reverse bias is due to avalanche breakdown. What makes them believe that this is avalanche breakdown? Especially, the rate of increase in the current shown in Fig. 2b appears too low for avalanche breakdown. If so, the authors should support their claim with:
 - A. Quantitative analysis for extracting impact ionization rate and carrier multiplication factor: many previous reports on avalanche breakdown phenomenon in 2D materials have shown much clear analyses based on the above
 - B. What is the extracted impact ionization rate of the homojunction and how does this compare with other 2D and conventional materials?
 - C. Breakdown voltage: determination of the breakdown voltage ($V_{\text{Breakdown}}$) is arbitrary and unscientific. The $V_{\text{Breakdown}}$ specified in Fig.2b and 2e appear to be similar to how one would extract turn-on voltage of FETs. Therefore, the plot shown in Fig.2e and the authors' claim for having a low $V_{\text{breakdown}}$ that corresponds to the threshold limit dictated by the bandgap is not justified.

D. The authors should have wondered whether the breakdown voltage will continuously go down below 240K. If this does, how can the authors explain such steep decrease in the breakdown voltage as T decreases? This is in contradiction to avalanche breakdown effect. Why should it go below the voltage that corresponds to the bandgap? It is highly unlikely that the bandgap of WSe₂ varies that much with T.

E. Qualitative evidence for avalanche breakdown: avalanche breakdown effect demonstrated in other 2D heterojunction (Nature Nanotech. 14, 2019) and 2D channels (ACS Nano 2019, Nano Research 2020, ACS Nano 2022) show much steeper increase in the current at the breakdown field by showing an abrupt increase in the current in I_d-V_d sweeps. The authors should re-read these papers and consider such systematic experimental methods.

3. Novelty: Disregarding their unjustified claim of 'approaching threshold limit', I can not find the novelty of the work in comparison to other studies that report avalanche breakdown in 2D heterojunctions and 2D channels. The authors should carefully reconsider the key message of the manuscript.

Reviewer #2 (Remarks to the Author):

The manuscript by Wang et al. presents an interesting study of low threshold avalanche effect where the threshold energy reaches the fundamental limit, leading a low loss carrier acceleration and multiplication. Many researchers have tried to create 2D avalanche photodetectors with low bias, high gain, and room temperature operation. In this work, it is noticeable that the team achieved the classical limit of the avalanche threshold to the energy gap level within the multiplication region of a van der Waals homojunction with a clean junction interface. However, this manuscript appears to be an incremental extension of their prior publication (Light Sci Appl 11, 170 (2022)), and this lack of novelty may not entirely align with the scope of Nature Communications. Moreover, I have several concerns and doubts that need to be addressed, as outlined in the following comments:

1. Authors claim that the low threshold behaviour of the avalanche is attributed to weak electron-phonon scattering. If the device is encapsulated with hBN, the disorder within the material will further decrease, resulting reduction the electron-phonon effect. Why not utilize a more optimal structure with hBN?

2. What is the specific reason for using WSe₂ instead of other TMDs? In principle, all 2D semiconductors can be used for multiplication region. According to Author's previous work (Light Sci Appl 11, 170 (2022)), it is said that p-n, p-p, and n-n junctions can be made just by combining different thicknesses.

3. In order to obtain the validity of the bidirectional-photovoltaic effect (BPV) and the operational mechanism of the author's avalanche device, it is necessary to show the materials used in this work, the reason for the thickness combination, and comparative data of device performance when using different combinations and junctions.

4. A Graphite/InSe Shottky photodetector (Adv.Mater. 34, 2206196 (2022)) has already shown that threshold breakdown voltage is close to the intrinsic limit. In this paper, a much higher avalanche gain and larger abrupt increase in I-V is accompanied by high on-current. What is different (advantages) from this paper in terms of fundamental part and performance?

5. Author mentioned about selective area dry-etching process to form the homojunction. The devices in this manuscript are made using randomly exfoliated flake and it is shown that this technology is not

used. It should show more detailed experimental process and results for better controllability of thickness combinations.

6. There should be large Schottky barrier between 2D channel and metal contacts. Author should explain how Schottky contacts in both few-layer WSe₂/Pt and Multi-layer WSe₂/Pt attributes the avalanche behaviours.

7. How much is the external quantum efficiency of device?

Reviewer #3 (Remarks to the Author):

Please see uploaded file.

[Editorial Note: Please see the next page for the report]

Reviewer Report (Manuscript # NCOMMS-23-23147)

In this manuscript, entitled “*Approaching the threshold limit of avalanche in stepwise van-der-Waals homojunctions*”, Wang et al reported the experimental investigation of avalanche photodiode based on van der Waals heterostructure (vdWH) composed of WSe₂. Avalanche photodiode is a practical component for detection of weak light signals (even down to few-photon level) and can be applied to a wide range of use cases, such as optical communication, imaging, remote sensing, and quantum applications. Conventional architecture of such devices, however, suffer multiple detrimental shortcomings, particularly the ultra-high threshold voltage (up to 200 V) that substantially increases the cost of design, fabrication, and energy consumption, thus impeding the practicality and sustainability aspects of avalanche photodiode. In this manuscript, the Authors have demonstrated an interesting alternative approach to reach the threshold limit of avalanche diode using a stepwise layered-material *homojunction* composed of WSe₂ of different layer number. The findings of low threshold voltage of 1.6 V at room temperature and bidirectional-photovoltaic behaviors are particularly interesting to me. Such effects are proposed to be associated with the adiabatic charge-carrier transport and impact ionization processes in the investigated stepwise homojunction. A positive temperature coefficient is also demonstrated, which provides clear evidence of the avalanche process.

I am convinced that the work represent a substantial step-forward (i.e. experimental synthesis of a novel device structure and demonstration of the underlying working principle) in avalanche diode technology. The findings appears to shed insights that can potentially motivate subsequent investigation, both device optimization (e.g. via different layered semiconductors, or metal contacts), and basic physics (e.g. the development of 2D avalanche theory for layered hetero/homostructures – note that the current simulation in Fig. 1b is based on ‘traditional’ TCAD software, which may not fully capture the physics of 2D layered materials). I am thus, overall, supportive for the manuscript to be accepted for publication, after appropriate minor revisions, in the **Nature Communications**.

I have also carefully gone through two articles, i.e. Adv. Mater. 2206196 (2022) and Light Sci. Appl. **11**, 170 (2022), which reports the avalanche photodiode based on InSe layered semiconductors and the proposal of layered semiconductor homojunction (i.e. layer “*PN diode*”), respectively. The Adv. Mater. 2206196 (2022) has reported an InSe/graphite heterojunction with 1.8 V threshold voltage – structurally different from the layer-number-based homojunction reported in this work. Despite both the Adv. Mater. and this manuscript reports similarly low threshold voltage, the low operation temperature and the negative temperature coefficient of Adv. Mater. suggest that the different device mechanisms – the room temperature compatibility of the homojunction reported in this manuscript may be more advantages than the InSe/graphite in terms of practical application. Although the Light Sci. Appl. **11**, 170 (2022) reported the avalanche-like behavior in a heavily doped PN junction, the threshold voltage (4 V) is rather high, and a low operation temperature is still needed. In this device, the (traditional) multiplication mechanism leads to high energy loss (proposed to be caused by the strong scattering processes in the channel with a high charge carrier concentration). I thus believe that this manuscript provides a substantially different, and potentially advantageous, avalanche photodiode architecture.

Below I provide some comments that the Authors should address so to improve the manuscript further.

1. As outlined in detail in my comment above, distinction between this work and previous works need to be more thoroughly discussed so to convincingly establish the appropriateness of this manuscript in Nat. Commun. The Authors could also consider to further supplement such discussion with a table that clearly contrast this work with other 2D-material-based and 'traditional' avalanche photodiode, in terms of mechanism, operating temperature, threshold voltage, photogain, dark current and so on (i.e. any other aspects that the Authors think could help to highlight the novelty of this manuscript) – perhaps some overlap with Fig. 3a, but I think such table will be immediately useful to establish the potential novelty of this work.
2. In Fig. 4, the distinct bidirectional-photovoltaic behaviors of the WSe₂ stepwise-layer-junction demonstrate the transition of the junction. It is an interesting phenomenon that is completely different from the common photovoltaic devices. However, how do the authors eliminate the possibility of non-ohmic contact formed on the two-dimensional WSe₂?
3. Minor point – why Fig. 4 middle panel use bar charts? Usually bar charts are more 'conventional' for histogram-like plot. Since it is an OC voltage vs laser plot, I suggest using the more conventional scatter plots format.
4. In relevance to the point 2 above, the presence of Schottky contact at metal/semiconductor interface is particularly problematic for 2D layered semiconductors [e.g. see Nat. Mater. 14, 1195 (2015)]. The corresponding Schottky barrier height can be characterized using the 2D semiconductor thermionic emission model [for example, see Phys. Rev. Lett. 121, 056802 (2018) and InfoMat 3, 502 (2021)] via an Arrhenius plot. Such contact barrier characterization may be useful in clearly illustrating the contact effects in their device – the Authors could consider performing such characterization OR if such measurement is not straightforward achievable in their device setup, the Authors could provide some brief discussions on this aspect, i.e. how Schottky contact could affect their proposed device operation and how the barrier height could be characterized.
5. In Fig. 2, the Authors demonstrated a good photoelectric performance of the proposed WSe₂-based avalanche photodiodes at room temperature. Some key parameters of photodetectors such as responsivity and response time are important for practical applications. The Authors are recommended to include such parameters – and/or any other device 'figure of merits' that will be useful for future works for benchmarking purposes.
6. The Authors should also comment on the 'duty cycle' aspect of their device – do they observe significant device degradation after measurement and overtime?
7. Some writing errors:
 - Pg. 5, "Those efforts reduce the avalanche voltage to ~10V @ 300K...", there should be a blank between "number" and "unit". Please check and modify similar errors in this

manuscript.

- Pg. 6, “A detailed discussion of such characteristic...”, the word “characteristic” should be “characteristics”, right?
- Pg. 7, “To provide further evidence of it, we performed temperature dependent dark current measurement on the...”, the phrase “temperature dependent dark current measurement” should be replaced by “temperature-dependent dark current measurements”.
- Pg. 10, “It might also be a clue to understand...”, the word “understand” should be “understanding”.

In summary, I recommend the manuscript to undergo a minor revision based on my points above. The manuscript can be accepted for publication in the **Nature Communications** after addressing my comments above.

Response to Reviewers' comments

We thank the referees for the constructive comments on the manuscript. We have addressed all the comments point-by-point and revised the manuscript accordingly. In this response letter, comments from referees are in black italic typeface, our responses are in a regular blue typeface. All major changes have been highlighted in blue in the main text and Supplementary Materials.

Comments from the reviewers:

Reviewer #1:

In this study, the authors present work on using 2D vdW homojunctions for observing an avalanche breakdown under a low voltage bias that is close to that of the threshold limit (i.e. electrical energy that corresponds to $\sim E_g$). Homojunction is composed of lateral few-layer/multi-layer junction naturally formed from mechanical exfoliation due to the different thicknesses of the exfoliated flake. The homojunction device is fabricated by depositing each Pt/Au electrode on each region (i.e. few-layer and multi-layer region of the flake). The authors attribute a significant increase in the current at a reverse-bias voltage to avalanche breakdown effect and discuss the breakdown voltage in terms of the onset of such large increase in the current. Some qualitative arguments are shown for explaining the temperature-dependence and the shift of the VOC according to the incident light intensity. In many aspects, I believe that many of the claims made by the authors in this manuscript are overstatements often lacking physical bases to clearly demonstrate avalanche breakdown in their 'homojunction' device. Therefore, I recommend rejection of this manuscript to as highly respected journal as Nature Communications and seriously recommend authors to thoroughly re-interpret their results. In details, I find their manuscript should not be submitted to another journal without re-interpreting their results without their unsubstantiated explanations based on avalanche breakdown and considering the points further below:

1. *Homojunction*: the authors should not use the word ‘homojunction’ for the device architecture that they propose; homojunctions typically consist of the same semiconductors but different doping levels (e.g. p-n junctions in Si). If they understand that the few-layer and multi-layer WSe₂ have different bandgaps due to their different electronic structure. It is also disappointing that the authors have not specified their relevant electronic structures; indirect or direct bandgap, which prompts the discussion of whether their claim of having ‘reduced el-ph coupling’ can be physically justified in the context of avalanche breakdown (which is not strictly proven, either).

Response: Thank you for your questions.

(1) **Reason for naming homojunction:** The device architecture proposed here differs from traditional homo- or hetero-junctions. On the one hand, it is fabricated on the same parent material, which thus can be identified as homo-junction. On the other hand, the bandgap transits from ~1.6 to ~1.2 eV as it crosses over from few-layer to multi-layer segments, which thus shows the signature of hetero-junction. In this regard, both homojunction and heterojunction are appropriate to describe the stepwise WSe₂ device.

But if to choose one, we would prefer to name it ‘homojunction’ rather than ‘heterojunction’. Because the 2D material community usually defines heterojunction as devices fabricated on two or more different materials, e. g. MoS₂/WSe₂, BP/MoS₂ (Nature 2020, 579, 368; Nat. Rev. Methods Primers 2022, 2, 58).

(2) **Band alignment of the proposed WSe₂ diode:** In the Supplementary Material, there is a separate section to discuss the band alignment and charge carrier transport (Section III, page 12). As shown in Figure R1c, the device shows a type I band structure, where the few-layer segment serves as a wide gap semiconductor (~1.6 eV), and the multilayer segment performs as a narrow gap counterpart (~1.2 eV). The SKPM (Scanning Kelvin Probe Microscopy) experiments help verify that the Fermi-level of the few-layer segment is much higher than that of the multi-layer part. Thus, in the junction region (Figure R1d), the few- and multi-layer WSe₂ bend upwards and downwards, respectively.

In such a device, there is a potential barrier in the conduction band, which denies the electron transport from multi-layer to few-layer direction (Figure R1d). For this reason, the multi-layer segment will contribute little to the photoresponse of the device. Such characteristic is confirmed by the SPCM (scanning photocurrent microscopy)

experiment. As displayed in Figure R2 (corresponding to Figure 3 in the main text), the characterized device is fabricated on a stepwise WSe₂ flake (6 L/55 L), with symmetric electrodes deposited on the few- and multi-layer segments, respectively. In such a device, the few-layer segment contributes the most response to the light illumination ($\lambda=520$ nm), while the multilayer part is blind to the illumination.

Figure R1. Energy band alignment and charge carrier transport in the stepwise layer junction. (a) AFM and KPFM images of a stepwise WSe₂ flake, where three distinct layer segments, 4 L, 9 L, and 14 L, are clearly labeled. (b) Corresponding height and potential profiles that are derived from a. (c) Band alignment of few- and multi-layer WSe₂ before contact. (d) Energy band diagram and the photocarrier transport process.

Figure R2. SPCM result of the stepwise WSe₂ diode.

(3) **Evidence of weak electron-phonon interaction:** Weak electron-phonon coupling is one of the intriguing properties of TMD materials. Although the underlying mechanism is still not completely understood, such characteristics have been demonstrated in previous works. In 2017 (Nat. Nanotechnol. 2017, 12, 1134), Fatemeh Barati et. al reported that the intrinsic weak electron-phonon coupling leads to a slow cool-down of hot carriers which helps trigger the interlayer (WSe₂/MoSe₂) multiplication. In 2019, Ji-Hee Kim et. al reported a 99% carrier-multiplication conversion efficiency in MoTe₂ and WSe₂ films (Nat. Commun. 2019, 10, 5488).

Our experiments also show evidence of such distinct properties. Figure R3c (corresponding to Figure 2 in the main text) shows the Raman spectra of 4 L and 39 L WSe₂ measured with a 532 nm laser line. The feature peaked at 249.3 cm⁻¹ is the first-order Raman signal, which arises from the A_{1g} phonon mode. Additional features peaked at 138.8, 257.5, 309.5, 360.6, 373.1, and 395.7 cm⁻¹ corresponding to the second-order Raman signals, that are associated with combination and overtones of phonons. Obviously, the second-order Raman signals decline as the thickness of WSe₂ scales from 39 L to 4 L, e. g. the 2LA(M) (peaked at 257.5 cm⁻¹) intensity declines from 67% to 24% of the A_{1g} intensity. It is consistent with the Raman results from previous reports (ACS Nano 2014, 8, 9, 9629–9635; Physical Review B, 2013, 88, 195313). Such character indicates that there are fewer phonons active in WSe₂ as its thickness shrinks to the monolayer limit. This might be a clue to understanding the intrinsic weak e-p interaction property of monolayers' thick TMD materials.

Figure R3. Device structure of Stepwise WSe₂ diode. (a) Schematically showing the device structure. (b) HAADF-TEM and EDX images of the stepwise WSe₂ structure. (c) Raman spectra of 4 L and 39 L WSe₂ measured with a 532 nm laser line. The spectra are normalized to the A_{1g} peak and vertically offset for clarity.

2. *Avalanche carrier multiplication mechanism: I can not find qualitative or quantitative supporting evidence for claiming that the observed increase of current under the reverse bias is due to avalanche breakdown. What makes them believe that this is avalanche breakdown? Especially, the rate of increase in the current shown in Fig. 2b appears too low for avalanche breakdown. If so, the authors should support their claim with:*

Response: Thank you very much for your questions.

(1) **Evidence of the avalanche breakdown:** For the judgment ‘*I can not find qualitative or quantitative supporting evidence for claiming that the observed increase of current under the reverse bias is due to avalanche breakdown*’, we have a different opinion. As is well documented in the textbook, the positive temperature coefficient ($\frac{\Delta V_b}{\Delta T} > 0$) is the most obvious characteristic of avalanche breakdown (IEEE Journal of Quantum Electronics 2010, 46, 1153; International Journal of Electronics 1972, 32, 23). The underlying physics is described as follows. As the temperature falls, electrons are gradually spared from phonons scatterings (by freezing the lattice vibration). Electrons then easily gain an excess energy of E_g and active the impact ionization process. For this reason, the electric voltage/power required for an avalanche is decreased continuously.

Figure R4. Dark I - V curves of WSe₂ diode at different temperatures.

In this work, we show that the temperature coefficient of the stepwise WSe₂ diode is positive. As shown in Figure R4 (corresponding to Figure 2e in the main text), the avalanche voltage gradually decreases as the temperature falls. More results on another two devices are shown in Figure R5, which further confirms that the avalanche process dominates the breakdown in stepwise WSe₂ diodes. We also fabricated several commercial InGaAs avalanche devices and show their temperature coefficient of them for comparison in Supplementary Section IV (Figure S12).

Figure R5. Dependence of breakdown voltage on the operating temperature for another two WSe₂ diodes.

(2) **Rate of increase in the current:** For the judgment ‘*the rate of increase in the current shown in Fig. 2b appears too low for avalanche breakdown.*’, we have a different opinion. To illustrate this issue, we compare the I - V curves of the WSe₂ device with those of commercial InGaAs avalanche diodes. As shown in Figure R6, both kinds of devices experience a $\sim 10^4$ times increase of current after avalanching. And, more importantly, their current climbs almost at the same rate, $\frac{dV}{d\log I} \approx 400 \text{ mV/dec}$.

Figure R6. Comparison of I - V curves between WSe_2 and InGaAs avalanche diodes in the dark. The inset: full-scale I - V curves of InGaAs avalanche diodes.

A. Quantitative analysis for extracting impact ionization rate and carrier multiplication factor: many previous reports on avalanche breakdown phenomenon in 2D materials have shown much clear analyses based on the above.

Response: Thank you very much for your questions.

Multiplication-factor: In the original manuscript, we have made a detailed discussion on the multiplication-factor/photo-gain of the stepwise WSe_2 diode. As depicted in Figure 2d (main text), the multiplication factor is retracted according to the equation, $M = \frac{I_{ph} - I_d}{I_{bg}}$ (Nat. Nanotech. 2019, 14, 217), where M is the multiplication factor, I_{ph} represents the photocurrent, I_d is the dark current and I_{bg} denotes the photocurrent when $M=1$.

We also made a comparison of the multiplication factor between bulk material and our WSe_2 diode. As shown in Figure R7, the multiplication factor of commercial Si and Ge avalanche diode is approximately 100, while that of our WSe_2 diode is up to 470.

Figure R7. Summary on the breakdown voltage and multiplication factor of WSe₂ and traditional avalanche diodes at room temperature. Note: each red solid circle represents an individual WSe₂ avalanche diode.

B. What is the extracted impact ionization rate of the homojunction and how does this compare with other 2D and conventional materials?

Response: Thank you very much for your questions.

Impact ionization rate: In the revised manuscript, we perform an additional discussion on the impact ionization rate of the stepwise WSe₂ diode. The impact ionization rate is derived by solving the equation, $1 - \frac{1}{M} = \int_0^W \alpha [\exp(\int_0^x -\alpha dx)] dx$, where M is the multiplication factor, α is the impact ionization rate, and W is the width of the channel or multiplication region (Physics of Semiconductor Devices, John Wiley Sons Inc., Hoboken, New Jersey, 2007). Considering that the avalanche process mainly arises from the hole impact ionization in stepwise WSe₂ diode, the rate calculated here is thus the hole impact ionization rate. Figure R8 summarizes the hole impact ionization rate of bulk material, uniform WSe₂, and our stepwise WSe₂ avalanche devices. One can clearly find that the bulk material requires an ultra-high uniform electric field, $5 \times 10^5 - 2 \times 10^6$ V/cm (corresponding to inverse electric field $0.5 \times 10^{-6} - 0.2 \times 10^{-5}$ cm/V), to raise the impact ionization rate to a level of $10^4 - 10^5$ cm⁻¹. In the uniform vdW WSe₂ materials, by contrast, the electric field required for an avalanche is lowered by

approximately 10 times. And in our stepwise WSe₂ avalanche devices, it is further reduced by 20 times, to an extremely low value.

Figure R8. Summary on the hole impact ionization rate of bulk material (derived from ACS Omega 2021, 6, 4574–4581), uniform WSe₂ (derived from ACS Nano 2022, 16, 5376–5383), and our stepwise WSe₂ avalanche devices.

C. Breakdown voltage: determination of the breakdown voltage ($V_{\text{Breakdown}}$) is arbitrary and unscientific. The $V_{\text{Breakdown}}$ specified in Fig.2b and 2e appear to be similar to how one would extract turn-on voltage of FETs. Therefore, the plot shown in Fig.2e and the authors' claim for having a low $V_{\text{breakdown}}$ that corresponds to the threshold limit dictated by the bandgap is not justified.

Response:

Thank you very much for your questions. Breakdown voltage is the threshold voltage at which the initiation of breakdown occurs (comprehensive materials processing. Newnes, 2014). It is easy to spot, e. g. by locating the onset point of dark current (Opt. Quant. Electron. 2015, 47, 1671). Following this rule, we determine the breakdown voltage of the WSe₂ diode as 1.44 V (Figure R9a).

Also, there are reports determining the break-down voltage by fitting the Miller express, $M(V) = \frac{1}{1 - (\frac{V}{V_b})^c}$, where M represents the photocurrent multiplication factor, V_b denotes the breakdown voltage, and c is a dimensionless constant (ACS Nano 2022,

16, 5376-5383). As depicted in Figure R9b, this method leads to a similar judgment on the break-down voltage, $V_b=1.65$ V.

Note that not only our experiments, but many previous literature prove that the V_b values derived from the dark current and photomultiplication are indistinguishable (IEEE Journal of Quantum Electronics 2010, 46, 1153-1157).

Based on the above, we cannot agree with the judgment made by the reviewer in this section.

Figure R9. (a) Determination of breakdown voltage by locating the onset point of dark current. (b) Determination of breakdown voltage by fitting the Miller express, $M(V) = \frac{1}{1-(\frac{V}{V_b})^c}$.

D. The authors should have wondered whether the breakdown voltage will continuously go down below 240K. If this does, how can the authors explain such steep decrease in the breakdown voltage as T decreases? This is in contradiction to avalanche breakdown effect. Why should it go below the voltage that corresponds to the bandgap? It is highly unlikely that the bandgap of WSe2 varies that much with T.

Response: Thank you very much for your questions.

For diode at avalanching, there are two voltage components contributing to the electron acceleration and subsequent impact ionization, external bias voltage and internal built-in potential. Generally, the external bias voltage is up to $100E_g/e$ in Si and GaN avalanche devices (E_g is the energy gap of the semiconductor; e is the elementary

charge), while the built-in potential is at a low level, $\sim 0.7E_g/e$. This makes people easily neglect the contribution of the latter.

In a low threshold avalanche device, however, the breakdown voltage is lowered to $\sim E_g/e$. In this regard, the contribution of built-in potential to the avalanche performance should not be neglected.

Figure R10. Calculated energy-band structure of WSe₂ diode at zero bias voltage (a) without and (b) with considering the discrepancy of doping characteristic between the few and multi-layer segments.

To clarify this kind of issue, we further perform a numerical simulation. The theoretical model is established with a commercial software package (Sentaurus-TCAD). It considers the stepwise geometry of the WSe₂ device, where the two distinct segments are set as 5.6 (8 L) and 20.3 nm (29 L) in thickness. The electron affinity and bandgaps are 3.7 and 1.6 eV, 4.0 and 1.2 eV, for few- and multi-layer parts, respectively (Opt. Quantum. Electron. 2020, 52, 1-14; Appl. Phys. Lett. 2013, 102, 012111).

As shown in Figure R10 and Table 1, the built-in potential arises from two aspects: the band-offset and the Fermi-level drop (induced by different doping polarity and concentration in the counterpart segments). According to the simulation, the band offset solely leads to an internal potential of 0.41 V (close to $\frac{1.6 \text{ eV} - 1.2 \text{ eV}}{e} = 0.4 \text{ V}$, Figure R9a), while the latter one rises to 1.18 V and even higher (Figure R9b). If we assume that there is no energy loss during the electron acceleration process, the minimum voltage required for the avalanche would be lowered to $\frac{E_g=1.6 \text{ eV}}{e} - 1.2 \text{ V} (1.52 \text{ V}) = 0.4 \text{ V} (0.08 \text{ V})$ (Table 1). This explains why the breakdown voltage could further

decrease as temperature drops.

Table 1. Summary on the built-in potential of WeSe₂ diode and minimum bias-voltage required for avalanche at low temperatures.

Minimum energy (required for avalanche)	Built-in potential		Minimum bias-voltage (required for avalanche) at low temperatures
	Band-offset	Fermi level drop	
$E_g, \sim 1.6 \text{ eV}$	$\sim 0.4 \text{ V}$	$0.5 E_g/e$	0.4 V
		$0.7 E_g/e$	0.08 V

E. Qualitative evidence for avalanche breakdown: avalanche breakdown effect demonstrated in other 2D heterojunction (Nature Nanotech. 14, 2019) and 2D channels (ACS Nano 2019, Nano Research 2020, ACS Nano 2022) show much steeper increase in the current at the breakdown field by showing an abrupt increase in the current in I_d - V_d sweeps. The authors should re-read these papers and consider such systematic experimental methods.

Response:

Thank you very much for your questions. To illustrate this issue, we compare the I - V curves of the WSe₂ device with those of commercial InGaAs avalanche diodes. As shown in Figure R6, both kinds of devices experience a $\sim 10^4$ times increase in the current after avalanching. And, more importantly, their current climbs almost at the same rate, $\frac{dV}{d \log I} \approx 400 \text{ mV/dec}$.

It is worth noting that 400 mV/dec is a standard rate of current avalanching. If one wants a faster increase of current flow, a quantum mechanism should be added to the avalanche process, as discussed in Nat. Nanotech. 2019, 14, 217-222. We also notice that some other works report fast current increase at avalanching (ACS Nano 2018, 12, 7109-7116; ACS Nano 2022, 16, 5376–5383; Nano Res. 2021, 14, 1961–1966). However, their I - V curves are all displayed in linear coordinates, which couldn't be taken as valid samples for comparison.

Based on those actualities, we cannot agree with the judgment made in this section.

3. Novelty: Disregarding their unjustified claim of ‘approaching threshold limit’, I can not find the novelty of the work in comparison to other studies that report avalanche breakdown in 2D heterojunctions and 2D channels. The authors should carefully reconsider the key message of the manuscript.

Response: Thank you for your valuable questions. The novelty of this work reflects on two aspects:

(1) We report on a van der Waals architecture that reaches the threshold limit of the avalanche at room temperature.

Figure R11. Summary on the room-temperature breakdown voltage of bulk and van der Waals materials. Data of the middle graph is reproduced from ACS Nano 2022, 16, 5376-5383.

Figure R11 and Supplementary Section VI (Table S3) summarize the room temperature breakdown voltage of bulk, vdW material, and our stepwise WSe₂ structure. Obviously, the threshold voltage is up to 150 V in bulk material. It poses severe requirements on material, operation, and signal processing. For instance, the driving and signal-processing circuit should be specially designed due to the ultra-high driving voltage (up to 150 V). These requirements dramatically increase the cost and thus limit the applications. Recently, inspired by the intriguing feature of vdW materials, avalanche devices based on uniform InSe, MoS₂, WSe₂ and BP flake, WSe₂/MoS₂, Graphite/InSe, and BP/InSe heterostructures have been investigated. Those efforts

reduce the avalanche voltage to $\sim 10V$ @ 300K, which stands for a big step forward. But note that the breakdown voltage is still out of the range of traditional digital circuits ($\pm 5V$ voltage range). In this work, we show that the stepwise WSe_2 architecture could further reduce the threshold voltage to 1.6 V (Figure 3a). It reaches the classic limit of avalanching since the external energy required for avalanching equals the energy gap of multiplication region, $V_{ex} * e \approx E_g = 1.6 eV$ (for bulk avalanche diode, the threshold energy must reach $22E_g$ and more.). This means that the thermodynamic loss is low during the charge carriers' acceleration process, which thus relaxes the restriction on the external electric power. As a return, the common digital circuits can be used to drive such avalanche diodes, which will greatly extend the application scenarios.

(2) We report on a unique bidirectional photovoltaic effect (BPV) in the van der Waals avalanche device.

Figure R12. BPV and PV effect. First row shows the BPV effect observed in the

stepwise WSe₂ diode. Second/third row shows the PV effect observed in the Schottky WSe₂ diode and commercial silicon diode. The three graphs, from left to right, depict the dependence of I - V curves, open circuit voltage, and net-photocurrent on illumination intensity ($\lambda=520$ nm), respectively.

For the Si diode and vdW Schottky diode, the open circuit voltage shows a monotonic relationship with the illumination intensity, from 0 to 0.32 V and 0 to 0.165 V, respectively (photovoltaic effect, PV, as shown in the second row of Figure R12). In the stepwise WSe₂ diode, by contrast, the open circuit voltage experiences a reversal, from +0.12 to -0.07 V, with an increasing light intensity (marked as the bidirectional photovoltaic effect, BPV, as shown in the first row of Figure R12). More results (derived from different devices, Supplementary Figure S20) confirm that this is a common behavior of stepwise layer junctions.

Reviewer #2:

The manuscript by Wang et al. presents an interesting study of low threshold avalanche effect where the threshold energy reaches the fundamental limit, leading a low loss carrier acceleration and multiplication. Many researchers have tried to create 2D avalanche photodetectors with low bias, high gain, and room temperature operation. In this work, it is noticeable that the team achieved the classical limit of the avalanche threshold to the energy gap level within the multiplication region of a van der Waals homojunction with a clean junction interface. However, this manuscript appears to be an incremental extension of their prior publication (Light Sci. Appl. 11, 170 (2022)), and this lack of novelty may not entirely align with the scope of Nature Communications. Moreover, I have several concerns and doubts that need to be addressed, as outlined in the following comments:

1. Authors claim that the low threshold behaviour of the avalanche is attributed to weak electron-phonon scattering. If the device is encapsulated with hBN, the disorder within the material will further decrease, resulting in a reduction of the electron-phonon effect. Why not utilize a more optimal structure with hBN?

Response:

Thank you for your valuable suggestions. Following your ideas, we encapsulate the WSe₂ diodes with hBN layers. Figure R1 shows the optical microscope images of such devices before and after each process. The device is composed of two BN layers sandwiching a WSe₂ homojunction. The WSe₂ and bottom hBN layers were first mechanically exfoliated onto individual Si/SiO₂ substrates. Then the WSe₂ layers are transferred onto the bottom hBN layer through a dry pick-up transfer technique. After that, standard lithography, electrode deposition, and lift-off processes were performed to ensure the electrical contact (Pt/Au, 20/70 nm). Finally, a top BN layer is transferred onto the WSe₂/BN layers to construct the hBN/WSe₂/hBN sandwich structure. All photoelectric measurements of hBN/WSe₂/hBN sandwich structure are performed at room temperature, the same experimental conditions as that of individual WSe₂ homojunction structures.

Figure R1. Schematic illustrating the fabrication process for the BN/WSe₂/BN sandwich structure. Optical microscope images of (a) WSe₂ and bottom hBN layers on the Si/SiO₂ substrates, (b) WSe₂/hBN structure obtained by a dry pick-up transfer technique, (c) WSe₂/BN photodiode fabricated by standard lithography and electrode deposition processes, and (d) hBN/WSe₂/hBN sandwich structure. The white, red, and blue dashed boxes represent the bottom hBN, WSe₂ homojunction, and top hBN respectively.

The I - V curves of four different devices in the dark and under illumination are shown in Figure R2. Like the bare WSe₂ homojunction devices, the encapsulated ones show significant avalanche behavior while negatively biased. The breakdown voltages are derived as -1.4, -1.5, -1.5, and -1.7 V, respectively, which is close to the theoretical limit, $E_g/e \approx 1.6$ V. Such characteristics further confirm that a low threshold avalanche is one of the intrinsic properties of the stepwise WSe₂ diodes.

Theoretically, the covering hBN layer could help reduce the surface recombination and thus lead to a lower threshold voltage as compared with the bare ones. At this moment, However, the experiment doesn't show obvious signs of this trend. This might be due to ① The limited number of samples: based on which we cannot draw firm conclusions. ② The intrinsically low surface recombination velocity: as is well documented, there is no surface dangling bond in the vdW materials, which thus leads to a weak surface effect. ③ Imperfect transfer process: residual glue can be easily introduced into the interface of WSe₂ and hBN during the physical transfer process, which will limit the performance improvement.

Figure R2. I - V curves of four hBN/WSe₂/hBN devices in the dark and under illumination at 300 K. Inset: optical microscope images of the devices before covering the top hBN layer.

2. What is the specific reason for using WSe₂ instead of other TMDs? In principle, all 2D semiconductors can be used for multiplication region. According to Author's previous work (*Light Sci. Appl.* 11, 170 (2022)), it is said that p - n , p - p , and n - n junctions can be made just by combining different thicknesses.

Response:

Thank you for your questions. We agree that other TMD materials, e. g. MoS₂ and WS₂, could be credible alternatives to realize avalanche functionality. In this work, we choose WSe₂ as the gain medium just because it is a well-known bipolar material, where the background doping concentration is extremely low, $\sim 10^8$ cm⁻², at least two orders lower than that of other TMD materials. In such a medium, the ionized impurities/defects scatterings are minimized, which would lead to a much-prolonged

minority carrier lifetime and thus higher quantum efficiency. A detailed discussion of the quantum efficiency of the WSe₂ avalanche device can be found in the Question 7 section.

Worth noting that this is a general rule in developing avalanche devices. Figure R10 shows the device structure of a commercial InGaAs avalanche diode (fabricated in this work for comparison). There is a dual impurity diffusion process, which forms a convex shape p-doping area. By this effort, the multiplication region is restricted to the unintentionally doped InP layer, rather than the heavily doped segments. In the Question 4 section, we will detailly discuss its I - V characteristic property and make a comparison with that of WSe₂ avalanche diodes.

Figure R3. Device structure of commercial InGaAs avalanche diode. (a) Top-view optical-microscope-image of the InGaAs avalanche pixel. (b) Schematically showing the dual impurity diffusion processes. (c) Scanning capacitance microscopy image taken at the cross-section of the device. The yellow and red color represents the p and n-type doping, respectively.

3. In order to obtain the validity of the bidirectional-photovoltaic effect (BPV) and the operational mechanism of the author's avalanche device, it is necessary to show the materials used in this work, the reason for the thickness combination, and comparative data of device performance when using different combinations and junctions.

Response: Thank you for your valuable questions.

(1) **Materials used in this work:** As schematically shown in Figure R4a, the stepwise WSe₂ flake was mechanically exfoliated onto a SiO₂/Si substrate in advance, and the electrical contacts were ensured by depositing Pt/Au electrodes on both sides. Figure R4b shows the high-angle-annular-dark-field scanning-transmission-electron-

microscope (HAADF-TEM) and energy-dispersive X-ray (EDX) images of the device by a Tecnai F20 TEM. The morphology transition between few and multilayer WSe₂ is atomically abrupt and the thickness of them is determined as 4 L and 39 L, respectively. We also fabricated another twenty devices based on different thickness combinations and carried out a statistical analysis of their photoelectric performances, as summarized in Table 1 and Figure R5-R10 (corresponding to Supplementary Section II). Figure R4c shows the Raman spectra of 4 L and 39 L WSe₂ measured with a 532 nm laser line by a Lab RAM HR 800. The feature peaked at 249.3 cm⁻¹ is the first-order Raman signal, which arises from the A_{1g} phonon mode. Additional features peaked at 138.8, 257.5, 309.5, 360.6, 373.1, and 395.7 cm⁻¹ corresponding to the second-order Raman signals, that are associated with combination and overtones of phonons. Obviously, the second-order Raman signals decline as the thickness of WSe₂ scales from 39 L to 4 L, e. g. the 2LA(M) (peaked at 257.5 cm⁻¹) intensity declines from 67% to 24% of the A_{1g} intensity. It is consistent with the Raman results from previous reports. Such character indicates that there are fewer phonons active in WSe₂ as its thickness shrinks to the monolayer limit. This might be a clue to understanding the intrinsic weak e-p interaction property of monolayers' thick TMD materials.

Figure R4. Device structure of Stepwise WSe₂ diode. (a) Schematically showing the device structure. (b) HAADF-TEM and EDX images of the stepwise WSe₂ structure. (c) Raman spectra of 4 L and 39 L WSe₂ measured with a 532 nm laser line. The spectra are normalized to the A_{1g} peak and vertically offset for clarity.

(2) **Reason for the thickness combination:** According to the reviewer's comments, we have summarized the thickness combination and breakdown voltage of twenty devices, as listed in Table 1. Figure R5a shows a statistical analysis of twenty WSe₂ diodes with different thickness combinations. Among these, devices with a breakdown voltage close to the threshold limit (~1.6 V) are marked by red balls. Several typical thickness combinations of eight WSe₂ diodes are selected, shown in Figure R5b. One can find that the few-layers part is below 8 L (6, 3, 8, 3, 7, 7, 8, and 3 L for eight

devices) and the multilayer part is over 17 L (29, 17, 25, 27, 33, 19, 30 and 55 L for eight devices). Detailed experimental data including optical microscope images, AFM images, height profiles extracted from AFM results, and dark I - V curves, of twenty WSe₂ diodes are shown in Figure R6-R10. Combined with these results, the thickness combination of few-/multi- layer should be 3-8 L/>20 L to realize an ideal performance.

Table 1. Summary of the WSe₂ diode with different thickness combinations.

Device Number (#)	Few-layer WSe ₂ (layer)	Multilayer WSe ₂ (layer)	Breakdown voltage (V)
1	~6	~29	-1.4
2	~5	~36	-2.1
3	~3	~17	-1.9
4	~4	~46	-5.0
5	~6	~28	-1.4
6	~8	~25	-1.7
7	~6	~40	-3.0
8	~5	~75	-1.9
9	~3	~16	-4.3
10	~3	~27	-1.6
11	~7	~33	-1.4
12	~13	~22	-3.0
13	~10	~53	-2.2
14	~3	~13	-5.0
15	~5	~54	-5.4
16	~7	~36	-4.8
17	~4	~45	-3.5
18	~3	~55	-1.4
19	~7	~19	-1.5
20	~8	~29	-1.5

Figure R5. (a) Statistical analysis of the stepwise WSe₂ diodes with different thickness combinations. (b) Typical WSe₂ diodes with a breakdown voltage close to the threshold limit (~ 1.6 V).

Figure R6. Optical microscope images, AFM images, height profiles extracted from AFM results, and dark I - V curves of Numbers #1, #5, #10, and #6 stepwise WSe₂ diodes.

Figure R7. Optical microscope images, AFM images, height profiles extracted from AFM results, and dark I - V curves of Numbers #11, #20, #19, and #18 stepwise WSe_2 diodes.

Figure R8. Optical microscope images, AFM images, height profiles extracted from AFM results, and dark I - V curves of Numbers #3, #2, #13, and #8 stepwise WSe_2 diodes.

Figure R9. Optical microscope images, AFM images, height profiles extracted from AFM results, and dark I - V curves of Numbers #7, #9, #12, and #17 stepwise WSe₂ diodes.

Figure R10. Optical microscope images, AFM images, height profiles extracted from AFM results, and dark I - V curves of Numbers #4, #15, #14, and #16 stepwise WSe₂ diodes.

(3) **Comparative data of device performance:** We have made a comparison of device performance among different thickness combinations. As depicted in Figure R11, there is indeed a strong dependency of bidirectional photovoltaic effect (BPV) on the thickness combination. Generally, the transition of open circuit voltage (from positive to negative value) requires a much higher illumination intensity as the thickness difference increases. It is responsible since the initial band-offset/built-in electric field is strong in those devices, e. g. Device 2, which needs more photo-carriers to reverse.

Figure R11. More data on bidirectional photovoltaic effect. The left column shows the optical microscope images of four different WSe_2 diodes, denoted as Device 1 (3 L/27 L), Device 2 (5 L/36 L), and Device 3 (8 L/50 L). The middle column shows the dark and photocurrent I - V curves of the four devices, respectively. The right column shows the dependence of open circuit voltage on the illumination intensity.

In this work, however, we haven't observed a strong dependence of thickness combinations on the avalanche breakdown voltage. As depicted in Figure R11, the break-down voltage of Device 1 (3 L/27 L), Device 2 (5 L/36 L), and Device 3 (8 L/50

L) are close to each other, 1.6 V, 1.8 V, and 2.1 V, respectively, despite the significant thickness variations. Such characteristics should arise from the distinct avalanche process of WSe₂ devices. Specifically, the charge-carrier acceleration and impact ionization mostly take place at the few-layer segments (see Figure 4a in the main text). The avalanche is then insensitive to the thickness variations of the devices, especially the variations in the thick segment.

Figure R12. Robustness of stepwise WSe₂ diodes in realizing the low threshold avalanche. Left to right: dark and photo-excited I - V curves of Device 1 (3 L/27 L), Device 2 (5 L/36 L), and Device 3 (8 L/50 L).

Given the facts above, stepwise WSe₂ diode is proved robust to realize the BPV and low-threshold avalanche effect, and we suggest that the thickness of the few-layer segment is 3-8 layers, while that of the thick part should ideally be more than 20 L.

4. *A Graphite/InSe Schottky photodetector (Adv.Mater. 34, 2206196 (2022)) has already shown that threshold breakdown voltage is close to the intrinsic limit. In this paper, a much higher avalanche gain and larger abrupt increase in I - V is accompanied by high on-current. What is different (advantages) from this paper in terms of fundamental part and performance?*

Response: Thank you for your valuable questions. The differences and advantages in this paper in terms of the working principle and performance can be summarized as follows:

(1) Discrepancy in the breakdown mechanism and operating temperature: We notice that Zhiyi Zhang et. al also reported a breakdown voltage close to the intrinsic limit (Adv. Mater. 2022, 34, 2206196). However, those devices can only work at low

temperatures, and more importantly, their temperature coefficients are negative, which is usually classified as Zener breakdown, rather than the avalanche process.

In this work, we show clear evidence of the avalanche process in stepwise WSe₂ diodes: a positive temperature coefficient. At the same time, the standard value of breakdown voltage (~1.6 V) is extracted at room temperature, consistent with the manner developed in bulk avalanche devices. Based on those solid facts, we state that the devices fabricated here reach the threshold limit of the avalanche.

(2) Avalanche gain: The high gain of the Graphite/InSe device relies on an extremely low operating temperature, and it degrades sharply as the temperature rises. As shown in Figure R13, the gain degrades from 10⁵ to 26 as the temperature rises from 100 to 200 K. Following this trend, the gain of the Graphite/InSe device is no better than that of Si and our WSe₂ avalanche devices at room temperature (100-220).

[REDACTED]

Figure R13. Gain as a relationship of temperature for Graphene/InSe avalanche photodetector (Adv. Mater. 2022, 34, 2206196), stepwise WSe₂ diode, and Si APD (Hamamatsu S12023 series; <https://www.hamamatsu.com.cn/cn/zh-cn/product/optical-sensors/apd/si-apd/S12023-02.html>).

(3) Larger abrupt increase in $I-V$: To illustrate this issue, we compare the IV curves of the WSe₂ device with that of commercial InGaAs avalanche diodes. As shown in Figure R14, both kinds of devices experience a ~10⁴ times increase of current after

avalanching. More importantly, their current climbs almost at the same rate, $\frac{dV}{d\log I} \approx 400\text{mV}/\text{dec}$. This is a standard rate of current avalanching.

Figure R14. Comparison of dark I - V curves between WSe_2 and InGaAs avalanche diodes. The inset: full-scale I - V curves of InGaAs avalanche diodes.

If one wants a faster increase in current flow, he should rely on the quantum process. Just like in a silicon transistor, the subthreshold swing is usually no less than 60 mV/dec. If in some specific cases, the subthreshold swing is reduced to ~ 20 mV/dec there must be a quantum tunneling process involved in the charge carrier transport. In fact, Zhiyi Zhang et al. state that their breakdown is attributed to quantum charge carrier transport assisting with gate voltage in an impact-ionization field-effect transistor.

(4) High on-current: High on-current is not welcome in avalanche photodetectors. On one hand, it leads to the overheating problem that usually burns down the devices. On the other hand, it results in high background noise/dark counts, which would lower the sensitivity of the devices.

For these reasons, the avalanche devices developed here are superior to their traditional counterpart, since the current before and after avalanche is several orders lower.

5. Author mentioned about selective area dry-etching process to form the homojunction. The devices in this manuscript are made using randomly exfoliated flake and it is shown that this technology is not used. It should show more detailed experimental process and results for better controllability of thickness combinations.

Response:

Thank you for your comments. We have demonstrated the controllability of thickness combinations for WSe₂ homojunction through reactive ion dry etching (RIE) technology. During RIE processes, plasma species including Ar and CF₄ are used for thinning the multilayer WSe₂ to few-layer WSe₂. The etch depth is fully controlled by modulating plasma density, pressure, temperature, and etch time. Figures R15, R16, R17, and R18 show the optical microscopy and atomic force microscopy images of stepwise WSe₂ devices fabricated in this way.

Figure R15. Schematic illustrating the preparation process for the WSe₂ diode by RIE method.

Figure R16. Controllability of thickness for WSe₂. Optical microscopy and atomic force microscopy images of the pristine and etched WSe₂ for (a) Sample 1 and (c) Sample 2. Height profiles along the dashed lines extracted from AFM images for (b) Sample 1 and (d) Sample 2.

Figure R17. Controllability of thickness for WSe₂. (a) Optical microscopy and atomic force microscopy images of the pristine and etched WSe₂. (b) Height profiles along the dashed lines extracted from AFM images.

We further characterize the avalanche performance of the RIE-etched WSe₂ diode. As shown in Figures R18a and b, the thick part is ~34 L as original, while the other segment is thinned to ~3 L through the selected-area-dry-etching process. Figure R18c shows the I - V curve of the as-prepared WSe₂ diode. One can find that such a device also shows significant avalanche breakdown at a low bias voltage (-6 V). It further confirms that the low threshold avalanche is the intrinsic property of the stepwise WSe₂ diode.

Worth noting that the breakdown voltage of the etched diode is somewhat higher than that of the mechanically exfoliated device. This should be associated with the crystal damages induced by the RIE process, which might increase the scattering events and thus lower the efficiency of the charge carrier accelerating. A full quantitative understanding of such characteristics requires further experimental and theoretical study.

Figure R18. Carrier multiplication characteristic of the WSe₂ diode fabricated by RIE etching. (a) Optical microscope and AFM image of the stepwise WSe₂ diode by RIE method. (b) Height profile showing the thickness of few-layer and multilayer WSe₂. (c) Dark I - V curve of the WSe₂ diode at 300 K.

6. There should be large Schottky barrier between 2D channel and metal contacts. Author should explain how Schottky contacts in both few-layer WSe₂/Pt and Multi-layer WSe₂/Pt attributes the avalanche behaviours.

Response:

Thank you for your valuable questions. To exclude the possible effect of metal-semiconductor contacts on avalanche, we performed the scanning-photocurrent-mapping (SPCM) measurements on the stepwise WSe₂ diode. SPCM is a non-destructive photoelectric characterization method that is commonly employed to confirm junction location, electrical-contacts configuration, and the origin of photoresponse (Science 2013, 340, 1311; Nat. Nanotechnol. 2008, 3, 486). The experimental setup is shown in Figure R19, where a focused laser spot (~1 μm in diameter) is controlled to scan over the WSe₂ diodes, and the generated photocurrent is recorded in real-time. In general, SPCM experiments provide an optical probe to test the photoresponse of each part of the WSe₂ diodes.

Figure R19. Schematic showing the SPCM measurement setup.

In the stepwise WSe₂ device, the work function of the Pt electrode ($\Phi_{\text{Pt}}=5.65$ eV) is close to that of WSe₂ ($\Phi_{\text{WSe}_2}=5.43$ eV). Thus, the effect of the Schottky barrier on the photocarrier harvest could be neglected. The SPCM results help to verify such judgment. As depicted in the left panel of Figure R20, no photoresponse is observed at the metal-WSe₂ interface, indicating a little Schottky barrier there.

We also show the SPCM pattern of the Schottky vdW device for comparison. The reference device is fabricated on a uniform WSe₂ flake, with Cr/Au electrodes deposited on both sides. Considering that the work function of Cr ($\Phi_{\text{Cr}}=4.5$ eV) is much lower

than that of WSe₂, there are large Schottky barriers and thus photoresponse at the metal-WSe₂ interface.

Figure R20. SPCM experiments on the stepwise WSe₂ diode and Schottky WSe₂ diode. The upper/bottom panels show the optical microscope and SPCM images of the two kinds of devices.

In the SPCM image of the WSe₂ diode (Figure R20, left panel), it is worth noting that the photocurrent tends to localize in the few-layer segment of the stepwise WSe₂ diode. Such character comes from the distinct band-alignment. As schematically shown in Figure R21, the energy band of the multi-/few-layer is bent downward/upward significantly. It gives rise to a potential barrier in the conduction band, which denies the electron transport from multi-layer to few-layer direction. For this reason, the multilayer segment contributes little to the photoresponse of the device.

Figure R21. (a) Band alignment of few- and multi-layer WSe₂ before contact. (b) Energy band diagram and the photocarrier transport process.

7. How much is the external quantum efficiency of device?

Response:

Thank you for your questions. The maximum external quantum efficiency (EQE) of the proposed WSe₂ diode is approximately 14.8% at 0 V and 2000% at 2 V, respectively. The detailed discussion is provided as follows.

Quantum efficiency (QE) is defined as the conversion efficiency of photon flux into electron flux, which determines the response of a detector to light. The quantum efficiency typically takes into account all kinds of external losses such as reflection and scattering and is often called EQE. It is defined as the ratio of extracted electrons (n_e) to incident photons (n_{total}), with a value between 0 and 100% unless carrier multiplication effects are present (Chem. Soc. Rev., 2015,44, 3691-3718). EQE can be calculated according to the responsivity (R):

$$EQE = \frac{n_e}{n_{total}} = R \frac{hc}{e\lambda}$$

Where h is Planck's constant, c is the speed of light, e is the electron charge, and λ is the incident wavelength. Responsivity can be written as $R = I_{ph}/P_{in}$, where I_{ph} is the net photocurrent and P_{in} is the incident light power illuminated on the photosensitive area of the device. Figure R22a shows the EQE as a function of incident power under the illumination of a 520 nm laser at 300 K. EQE is improved with the growing biases from 14.8% (maximum value at 0 V) to 2000% (maximum value at 2 V). It originated from

the high photogain induced by carrier multiplication effects at $V_{ds}=2$ V. It is worth noting that there is a turning point for EQE with the growing power, which is caused by the bidirectional-photovoltaic effect. We have also measured wavelength-dependent EQE at zero bias, as shown in Figure R22b. One can find that the EQE is greater than 10% spanning a waveband from 500 to 800 nm, corresponding to the bandgap of few-layer WSe₂ (1.6 eV).

Figure R22. (a) External quantum efficiency as a function of incident power with increasing bias voltages. (b) Wavelength-dependent external quantum efficiency at zero bias.

Reviewer #3:

In this manuscript, entitled “Approaching the threshold limit of avalanche in stepwise van-der-Waals homojunctions”, Wang et al reported the experimental investigation of avalanche photodiode based on van der Waals heterostructure (vdWH) composed of WSe₂. Avalanche photodiode is a practical component for detection of weak light signals (even down to few-photon level) and can be applied to a wide range of use cases, such as optical communication, imaging, remote sensing, and quantum applications. Conventional architecture of such devices, however, suffer multiple detrimental shortcomings, particularly the ultra-high threshold voltage (up to 200 V) that substantially increases the cost of design, fabrication, and energy consumption, thus impeding the practicality and sustainability aspects of avalanche photodiode. In this manuscript, the Authors have demonstrated an interesting alternative approach to reach the threshold limit of avalanche diode using a stepwise layered-material homojunction composed of WSe₂ of different layer number. The findings of low threshold voltage of 1.6 V at room temperature and bidirectional-photovoltaic behaviors are particularly interesting to me. Such effects are proposed to be associated with the adiabatic charge-carrier transport and impact ionization processes in the investigated stepwise homojunction. A positive temperature coefficient is also demonstrated, which provides clear evidence of the avalanche process.

I am convinced that the work represent a substantial step-forward (i.e. experimental synthesis of a novel device structure and demonstration of the underlying working principle) in avalanche diode technology. The findings appears to shed insights that can potentially motivate subsequent investigation, both device optimization (e.g. via different layered semiconductors, or metal contacts), and basic physics (e.g. the development of 2D avalanche theory for layered hetero/homostructures – note that the current simulation in Fig. 1b is based on ‘traditional’ TCAD software, which may not

fully capture the physics of 2D layered materials). I am thus, overall, supportive for the manuscript to be accepted for publication, after appropriate minor revisions, in the Nature Communications.

I have also carefully gone through two articles, i.e. Adv. Mater. 2206196 (2022) and Light Sci. Appl. 11, 170 (2022), which reports the avalanche photodiode based on InSe layered semiconductors and the proposal of layered semiconductor homojunction

(i.e. layer “PN diode”), respectively. The *Adv. Mater.* 2206196 (2022) has reported an InSe/graphite heterojunction with 1.8 V threshold voltage – structurally different from the layer-number-based homojunction reported in this work. Despite both the *Adv. Mater.* and this manuscript reports similarly low threshold voltage, the low operation temperature and the negative temperature coefficient of *Adv. Mater.* suggest that the different device mechanisms – the room temperature compatibility of the homojunction reported in this manuscript may be more advantages than the InSe/graphite in terms of practical application.

Although the *Light Sci. Appl.* 11, 170 (2022) reported the avalanche-like behavior in a heavily doped PN junction, the threshold voltage (4 V) is rather high, and a low operation temperature is still needed. In this device, the (traditional) multiplication mechanism leads to high energy loss (proposed to be caused by the strong scattering processes in the channel with a high charge carrier concentration). I thus believe that this manuscript provides a substantially different, and potentially advantageous, avalanche photodiode architecture.

Below I provide some comments that the Authors should address so to improve the manuscript further.

1. As outlined in detail in my comment above, distinction between this work and previous works need to be more thoroughly discussed so to convincingly establish the appropriateness of this manuscript in *Nat. Commun.* The Authors could also consider to further supplement such discussion with a table that clearly contrast this work with other 2D-material-based and ‘traditional’ avalanche photodiode, in terms of mechanism, operating temperature, threshold voltage, photogain, dark current and so on (i.e. any other aspects that the Authors think could help to highlight the novelty of this manuscript) – perhaps some overlap with Fig. 3a, but I think such table will be immediately useful to establish the potential novelty of this work.

Response:

Thank you for your valuable comments. It is important to help us improve the quality of this work. According to your suggestions, we have summarized the performance metrics of several typical avalanche photodetectors (APDs) based on bulk materials and two-dimensional layered materials, as shown in Table 1. One can find that APDs based on conventional bulk materials show good performance at room

temperature, but often require high breakdown voltage (more than 25 V). Besides, all bulk materials-based APDs show a positive temperature coefficient, which is a typical feature of the avalanche multiplication process. In comparison, APDs based on two-dimensional materials have relatively low breakdown voltage due to their small size but often operate at low-temperature conditions. Our work proposes a WSe₂ stepwise homojunction diode with a low avalanche threshold voltage of about 1.6 V, a low dark current of about 10 fA, and a large avalanche gain of about 200 at room temperature. Importantly, the proposed WSe₂ diodes demonstrate a positive temperature coefficient, the same as that of the traditional bulk APDs.

Table 1. Comparison of performance metrics for different types of APDs

	APD structure	Operating temperature	Temperature coefficient	Breakdown voltage	Gain	Dark current	Ref.
Bulk material	Si PIN junction	300 K	Positive	-150 V	~100 @-150 V	~50 pA	1
	InGaAs PIN junction	300 K	Positive	-55 V	~30 @-55 V	~40 nA	2
	Si-Ge PIN junction	300 K	Positive	-25 V	~30 @-25 V	~100 pA	3
	GaN PIN junction	300 K	Positive	-92 V	~300 @-92 V	~100 pA	4
	Ge PIN junction	300 K	Positive	-30 V	~100 @-30 V	~300 nA	5
	AlGaIn Schottky junction	-	-	-50 V	1560 @-68 V	~10 pA	6
Two-dimensional material	WSe ₂ /WS ₂ PN heterojunction	-	-	-8.5 V	~300 @-16.5 V	~100 pA	7
	MoS ₂ /WSe ₂ PN heterojunction	300 K	-	-6.5 V	~5 @-10 V	~100 pA	8
	MoS ₂ PN homojunction	100 K	-	-4.5 V	~100 @-10 V	~1 pA	9
	Gr/InSe Schottky junction	260 K	negative	-1.8 V	~110 @-1.8 V	~1 pA	10
	BP/InSe PN heterojunction	80 K	negative	-4.8 V	~100 @-2 V	~100 pA	11
	WSe₂ Stepwise homojunction	300 K	Positive	-1.6 V	~200 @-2 V	~10 fA	This work

[1] Hamamatsu S12023 series, etc. <https://www.hamamatsu.com.cn/cn/zh-cn/product/optical-sensors/apd/si-apd/S12023-02.html>

[2] Hamamatsu G8931 series. <https://www.hamamatsu.com.cn/cn/zh->

cn/product/optical-sensors/apd/ingaas-apd/G8931-04.html

- [3] Kang Y, Liu H, Morse M, et al. Monolithic germanium/silicon avalanche photodiodes with 340 GHz gain–bandwidth product. *Nature Photonics*, 2009, 3, 59–63.
- [4] Verghese S, McIntosh K A, Molnar R J, et al. GaN avalanche photodiodes operating in linear-gain mode and Geiger mode. *IEEE Transactions on Electron Devices*, 2001, 48, 502-511.
- [5] <https://www.ushio.com/product/pd-ld-germanium-avalanche-photodiode/>.
- [6] Tut T, Gokkavas M, Inal A, et al. Al_xGa_{1-x}N-based avalanche photodiodes with high reproducible avalanche gain. *Applied Physics Letters*, 2007, 90, 163506.
- [7] Meng L, Zhang N, Yang M, et al. Low-voltage and high-gain WSe₂ avalanche phototransistor with an out-of-plane WSe₂/WS₂ heterojunction. *Nano Research*, 2023, 16, 3422-3428.
- [8] Son B, Wang Y, Luo M, et al. Efficient Avalanche Photodiodes with a WSe₂/MoS₂ Heterostructure via Two-Photon Absorption. *Nano letters*, 2022, 22, 9516-9522.
- [9] Xia H, Luo M, Wang W, et al. Pristine PN junction toward atomic layer devices. *Light: Science & Applications*, 2022, 11, 170.
- [10] Zhang Z, Cheng B, Lim J, et al. Approaching the intrinsic threshold breakdown voltage and ultrahigh gain in a graphite/InSe Schottky photodetector. *Advanced Materials*, 2022, 34, 2206196.
- [11] Gao A, Lai J, Wang Y, et al. Observation of ballistic avalanche phenomena in nanoscale vertical InSe/BP heterostructures. *Nature Nanotechnology*, 2019, 14, 217-222.

Based on Table 1, we further show the breakdown voltage and gain of Ge, Si, Ge-Si, InGaAs, GaN, AlGaIn diode, and our stepwise WSe₂ devices, as shown in Figure R1a. Obviously, the threshold voltage is up to 150 V in traditional avalanche architecture. It poses severe requirements on material, operation, and signal processing. For instance, there are few choices of materials to fabricate such kinds of devices, since they must be clean enough (background doping concentration in $\sim 10^{15} \text{ cm}^{-3}$) to bear the extremely high electric field ($0.1\text{-}1 \text{ MV/cm}^2$) without destructive breakdown. Meanwhile, the driving and signal-processing circuit should be specially designed due to the ultra-high driving voltage (up to 150 V). These requirements dramatically increase the cost and thus the applications. Yet now the TMD junction reduces the threshold value to 1.6 V. It reaches the classic limit of avalanching since the external

energy required for avalanching equals the energy gap of multiplication region, $V_{ex} * e \approx E_g = 1.6 \text{ eV}$ (for bulk avalanche diode, the threshold energy must reach $22E_g$ and more.). This means that the thermodynamic loss is low during the charge carriers' acceleration process, which thus relaxes the restriction on the external electric power. As a return, the common digital circuits ($\pm 5 \text{ V}$ voltage range) can be used to drive such avalanche diodes, which will greatly extend the application scenarios.

Figure R1. (a) Summary of the breakdown voltage and multiplication factor for WSe₂ and traditional avalanche diodes at room temperature. Note: each red solid circle represents an individual WSe₂ avalanche diode. (b) Summary on the hole impact ionization rate of bulk material (derived from ACS Omega 2021, 6, 4574–4581), uniform WSe₂ (derived from ACS Nano 2022, 16, 5376–5383), and our stepwise WSe₂ avalanche devices.

We further calculate the impact ionization rate of the stepwise WSe₂ device, by solving the equation, $1 - \frac{1}{M} = \int_0^W \alpha [\exp(\int_0^x -\alpha dx)] dx$, where M is the multiplication factor, α is the impact ionization rate, and W is the width of the channel or multiplication region. Considering that the avalanche process mainly arises from the hole impact ionization in stepwise WSe₂ diode, the rate calculated here is thus the hole impact ionization rate. Figure R1b summarizes the hole impact ionization rate of bulk material, uniform WSe₂, and our stepwise WSe₂ avalanche devices. One can clearly find that the bulk material requires an ultra-high uniform electric field, $5 \times 10^5 - 2 \times 10^6 \text{ V/cm}$ (corresponding to inverse electric field $0.5 \times 10^{-6} - 0.2 \times 10^{-5} \text{ cm/V}$), to raise the impact ionization rate to a level of $10^4 - 10^5 \text{ cm}^{-1}$. In the uniform vdW WSe₂ materials, by contrast, the electric field required for an avalanche is lowered by approximately 10 times. And in our stepwise WSe₂ avalanche devices, it is further reduced by 20 times, to an extremely low value.

2. In Fig. 4, the distinct bidirectional-photovoltaic behaviors of the WSe₂ stepwise-layer-junction demonstrate the transition of the junction. It is an interesting phenomenon that is completely different from the common photovoltaic devices. However, how do the authors eliminate the possibility of non-ohmic contact formed on the two-dimensional WSe₂?

Response:

Thank you for your valuable questions. To exclude the possible effect of metal-semiconductor contacts on the BPV phenomenon, we have performed the scanning-photocurrent-mapping (SPCM) measurements on the stepwise WSe₂ diode. SPCM is a non-disruptive photoelectric characterization method that is commonly employed to confirm junction location, electrical-contacts configuration, and the origin of photoresponse (Science 2013, 340, 1311; Nat. Nanotechnol. 2008, 3, 486). The experimental setup is shown in Figure R2, where a focused laser spot (<1 μm in diameter) is controlled to scan over the WSe₂ diodes and the generated photocurrent is recorded in real-time. In general, SPCM experiments provide an optical probe to test the photoresponse of each part of the WSe₂ diodes.

Figure R2. Schematic showing the SPCM measurement setup.

In such a WSe₂ device, the work function of the Pt electrode ($\Phi_{\text{Pt}}=5.65$ eV) is close to that of WSe₂ ($\Phi(\text{WSe}_2)=5.43$ eV). Thus, the effect of the Schottky barrier on the photocarrier harvest could be neglected. The SPCM results help to verify such judgment. As depicted in the left panel of Figure R3, no photoresponse is observed at the metal-WSe₂ interface, indicating a little Schottky barrier there. We also show the SPCM

pattern of the Schottky vdW device for comparison. The reference device is fabricated on a uniform WSe₂ flake, with Cr/Au electrodes deposited on both sides. Considering that the work function of Cr ($\Phi_{\text{Cr}}=4.5$ eV) is much lower than that of WSe₂, there are large Schottky barriers and thus photoresponse at the metal-WSe₂ interface.

Figure R3. SPCM experiments on the stepwise WSe₂ diode and Schottky WSe₂ diode. The upper/lower panels show the optical-microscope and SPCM images of the two kinds of devices.

In the SPCM image of the WSe₂ diode (Figure R3, left panel), it is worth noting that the photocurrent tends to localize in the few-layer segment of the stepwise WSe₂ diode. Such character comes from the distinct band-alignment. As schematically shown in Figure R4, the energy band of the multi-/few-layer is bent downward/upward significantly. It gives rise to a potential barrier in the conduction band, which denies the electron transport from multi-layer to few-layer direction. For this reason, the multi-layer segment contributes little to the photoresponse of the device.

Figure R4 (a) The Band alignment of few- and multi-layer WSe₂ before contact. (b) Energy band diagram and the photocarrier transport process.

We further perform the measurements of I - V curves for stepwise WSe₂ diode and Schottky WSe₂ diode, as shown in Figure R5. For the stepwise WSe₂ diode (Figure R5, upper panel), the open circuit voltage experiences a reversal (bidirectional-photovoltaic effect), from +0.12 to -0.07V, with increasing light intensity. In contrast, the open circuit voltage shows a monotonic relationship with the illumination intensity (photovoltaic effect), from 0 to 0.165 V for the Schottky WSe₂ diode, as shown in the bottom panel of Figure R5. Finally, those features help to rule out the possible effect of Schottky barriers on the photoresponse of the stepwise WSe₂ diode.

Figure R5. BPV and PV effect. First row shows the BPV effect observed in the stepwise WSe₂ diode. Second/third row shows the PV effect observed in the

Schottky WSe₂ diode. The three graphs, from left to right, depict the dependence of I - V curves, open circuit voltage, and net-photocurrent on illumination intensity ($\lambda=520$ nm), respectively.

3. *Minor point – why Fig. 4 middle panel use bar charts? Usually bar charts are more ‘conventional’ for histogram-like plot. Since it is an OC voltage vs laser plot, I suggest using the more conventional scatter plots format.*

Response:

Thank you for your valuable suggestion. We have modified it and displayed the OC voltage vs laser plot in scatter plot format, shown in Figure 3 in the main text.

4. *In relevance to the point 2 above, the presence of Schottky contact at metal/semiconductor interface is particularly problematic for 2D layered semiconductors [e.g. see Nat. Mater. 14, 1195 (2015)]. The corresponding Schottky barrier height can be characterized using the 2D semiconductor thermionic emission model [for example, see Phys. Rev. Lett. 121, 056802 (2018) and InfoMat 3, 502 (2021)] via an Arrhenius plot. Such contact barrier characterization may be useful in clearly illustrating the contact effects in their device – the Authors could consider performing such characterization OR if such measurement is not straightforward achievable in their device setup, the Authors could provide some provide brief discussions on this aspect, i.e. how Schottky contact could affect their proposed device operation and how the barrier height could be characterized.*

Response:

Thank you very much for providing this valuable information. The interface quality and contact barrier are important for the electrical and photoelectric performance in devices based on two-dimensional (2D) semiconductors with an atomic-layer thickness. In our work, the contact barrier between 2D WSe₂ and Pt/Au metal electrodes is critical to classify the origin of avalanche multiplication and bidirectional-photovoltaic behaviors. We have carefully gone through these articles (Phys. Rev. Lett. 2018, 121, 056802 and InfoMat 2021, 3, 502), which reported a feasible method to obtain the Schottky barrier height by thermionic emission model. These two papers are certainly very important for characterizing the Schottky barrier

between metal electrodes and semiconductors. After fully understanding the contact barrier characterization method in these articles and considering the device structure in our work, however, we don't think it is completely suitable for our work.

To make it more straightforward, we further performed Scanning Kelvin probe microscopy (SKPM) and scanning photocurrent mapping (SPCM) measurements on the stepwise WSe₂ diode. Those results help to verify that the Schottky barrier is extremely low in our devices and its effect on avalanche and BPV effect could be neglected.

We first carried out the SKPM experiments on the device (Figure R6). By this effort, the contact potential difference between Pt/Au and WSe₂ is derived, ~55 mV. Such a tiny potential drop arises from the low contrast of work function between WSe₂ ($\phi_{WSe_2} = 5.43 \text{ eV}$) and Au/Pt ($\phi_{Pt} = 5.65 \text{ eV}$, $\phi_{Au} = 5.1 \text{ eV}$). It indicates that there is little Schottky barrier located at the metal-semiconductor interface. Such judgment is also supported by the SPCM experiments, where no photoresponse is observed at the metal-semiconductor interface (Figure R3).

Following your suggestions, we add more experimental data and give further discussion with these articles (cited as Ref. [30] and [31]) in the revised manuscript.

Figure R6. (a) Optical microscope image of a WSe₂ homojunction diode. (c) Atomic force microscope images near the interface of (b) WSe₂ homojunction and (c) WSe₂/electrode. (d) Scanning Kelvin probe microscopy image of WSe₂ and Au electrode. (e) Height and potential profiles across the WSe₂/Au interface showing the contact potential difference between WSe₂ and Au electrode.

5. In Fig. 2, the Authors demonstrated a good photoelectric performance of the proposed WSe₂ - based avalanche photodiodes at room temperature. Some key parameters of photodetectors such as responsivity and response time are important for practical applications. The Authors are recommended to include such parameters – and/or any other device ‘figure of merits’ that will be useful for future works for benchmarking purposes.

Response:

Thank you very much for your suggestions. Responsivity, response time, external quantum efficiency, etc. are important performance parameters for determining the detection capability and practical application scenario of the detector. We have performed detailed measurements of the proposed WSe₂ diode under different light powers ranging from 500 to 1000 nm at room temperature.

Figure R7. (a) Responsivity as a function of incident power with increasing bias voltages. (b) Wavelength-dependent responsivity at zero bias.

(1) **Responsivity (R)** is the ratio between the net photocurrent (I_{ph}) and the total incident optical power (P) on the photodetector, which can be calculated by $R= I_{ph}/P$. Figure R7a shows R as a function of incident power under the illumination of 520 nm laser at 300 K. R can be improved by increasing applied voltages and reach approximately 10 A/W at -2 V. It originated from the high photogain induced by carrier multiplication effects at 2 V (above the avalanche threshold voltage). It is worth noting that there is a turning point for R with the growing power at zero bias, which is caused by the bidirectional-photovoltaic effect. Figure R7b shows wavelength-dependent responsivity at zero bias. One can find that the R is about 0.1 A/W spanning a waveband

from 500 to 800 nm, corresponding to the bandgap of a few-layer WSe₂ (1.6 eV).

(2) **External quantum efficiency (EQE)** is defined as the ratio of extracted electrons (n_e) to incident photons (n_{total}), with a value between 0 and 100% unless carrier multiplication effects are present (Chem. Soc. Rev., 2015,44, 3691-3718). EQE can be calculated according to the R :

$$EQE = \frac{n_e}{n_{total}} = R \frac{hc}{e\lambda}$$

Where h is Planck's constant, c is the speed of light, e is the electron charge, and λ is the incident wavelength. The calculated EQE as a function of light power with increasing bias voltages is shown in Figure R8a. The maximum EQE is 14.8% and 2000% at 0 and 2 V, respectively. We have also measured wavelength-dependent EQE at zero bias, as shown in Figure R8b. One can find that the EQE is greater than 10% spanning a waveband from 500 to 800 nm, corresponding to the bandgap of few-layer WSe₂ (1.6 eV).

Figure R8. (a) External quantum efficiency as a function of incident power with increasing bias voltages. (b) Wavelength-dependent external quantum efficiency at zero bias.

(3) **Response time:** the rising/falling time is defined as the total time needed for the photocurrent to rise (fall) from 10% (90%) to 90% (10%) of the peak. The falling and rising time of the photodetector usually depends on the carrier generation time, transport time, recombination time, and the external circuit time constant. The fast response speed of the homojunction photodetector is believed to be the quick separation and transportation of photogenerated carriers by the built-in electric field at the junction interface and the shorter lifetime of the carrier. Figure R9 shows the rising/falling time

of the proposed WSe₂ diode at 0 and -3 V. One can find that the rising/falling increased from 70/85 μs to 10/9 μs with the growing bias voltages. Worth noting that the response time could be limited by the RC time constant of the measurement system.

Figure R9. Time-resolved photoresponse of the device under 520 nm laser illumination at (a) $V_{\text{ds}} = 0 \text{ V}$ and (b) -3 V .

(4) Actual imaging application: To demonstrate the practical feasibility of the WSe₂ photodiode with superior photoelectric performance, we further performed the actual imaging measurements by a homemade imaging system at room temperature and ambient conditions. The main components of the imaging system are composed of a photodetector, an optical system (including lens, filter, and so on), a two-dimensional motor controller, and a current preamplifier. The prepared WSe₂ stepwise junction photodiode was integrated into the imaging system as the detector and operated at $V_{\text{ds}}=0 \text{ V}$. As shown in Figure R10a, a SITP hollow graphic under white light illumination is used as the imaging target. The light of the target was focused on the detector through the optical system. The motor controller was used to drive the detector to scan the whole

target in two-dimensional pixel by pixel. The generated photocurrent signals of each pixel were then amplified by the preamplifier and recorded to obtain a two-dimensional image. Figure R10a (upper right panel) shows the imaging result of the SITP letters. One can find that a clear SITP pattern with sharp boundaries is obtained. We further extract the photocurrent profile along the horizontal direction of the target, shown in Figure S10b. The photocurrent profile shows an obvious increase with the position movement from the background to the pattern body, which indicates the growing optical power from the background to the pattern edge and pattern body. The superior photoelectric performance of our WSe₂ photodiodes is confirmed by successfully identifying the difference between very high- and low-brightness targets and obtaining a clear imaging result.

Figure R10. Imaging application of the WSe₂ photodiode. (a) Schematic diagram of the single-pixel scanning imaging system. SITP (the abbreviation of Shanghai Institute of Technology and Physics) letter graphics under white light illumination are used as imaging targets. (b) Normalized photocurrent profile along the horizontal direction of the target (marked by red dotted lines in imaging photographs).

6. The Authors should also comment on the ‘duty cycle’ aspect of their device – do they observe significant device degradation after measurement and overtime?

Response: Thank you for your valuable questions. Stability and repeatability are very important for the practical application of the photoelectric device. We have demonstrated the stability testing of the proposed WSe₂ diodes to confirm the device degradation after measurement and overtime at room temperature, as shown in Figure R11. One can find that the breakdown voltages are almost unchanged after 50 consecutive tests, remaining at 1.6 and 2 V for Device ① and ②.

Figure R11. Dark I - V curves of 50 consecutive tests for WSe₂ diodes (a) Device ① and (b) Device ②.

To verify the stability of the WSe₂ diodes under laser illumination, we also measure the photoswitching response at zero bias under a periodical on/off 520 nm laser illumination at room temperature. The current rapidly increases when the laser is on and recovers to the dark state quickly when it is off. No obvious baseline drift in time-resolved photocurrent is observed with >200 cycles of modulated laser illumination, which confirms the good photoswitching stability of the devices. Considering the above photoelectric characterization results, we think the proposed WSe₂ photodiode exhibits good electrical and photoelectric stability and repeatability.

Figure R12. Photoswitching response of the WSe₂ diode under the illumination of 520 nm at $V_{ds}=0$ V.

7. Some writing errors:

- Pg. 5, “Those efforts reduce the avalanche voltage to $\sim 10V @ 300K...$ ”, there should be a blank between “number” and “unit”. Please check and modify similar errors in this manuscript.
- Pg. 6, “A detailed discussion of such characteristic...”, the word “characteristic” should be “characteristics”, right?
- Pg. 7, “To provide further evidence of it, we performed temperature dependent dark current measurement on the...”, the phrase “temperature dependent dark current measurement” should be replaced by “temperature-dependent dark current measurements”.
- Pg. 10, “It might also be a clue to understand...”, the word “understand” should be “understanding”.

In summary, I recommend the manuscript to undergo a minor revision based on my points above. The manuscript can be accepted for publication in the Nature Communications after addressing my comments above.

Response:

Thank you for pointing out these errors. We have modified these errors and carefully went through the English writing in the revised manuscript.

Revised:

- 1) Page 5: It has been modified to be “Those efforts reduce the avalanche voltage to ~10 V@300 K, which stands for a big step forward.” on Page 13.
- 2) Page 6: It has been modified to be “A detailed discussion of such characteristics can be found in Figure 5 and Supplementary Section III.” on Page 7.
- 3) Page 7: It has been modified to be “We has we further performed variant temperature experiments on several stepwise WSe2 diodes.” on Page 9.
- 4) Page 10: It has been modified to be “This might be a clue to understanding the intrinsic weak e-p interaction property of TMD materials.” on Page 7.

Based on the reviewers' comments, we have performed the following corrections:

Revised Manuscript:

1. Page 2 and 3, we added a discussion of the technical advantages of our stepwise diode in the abstract and introduction.
2. Page 4, we reorganized Figure 1 and revised the figure captions.
3. Page 4 and 5, we added a discussion of electron-photon scattering in two-dimensional TMD materials.
4. Page 6, we added a discussion of the difference in multiplication processes between bulk and TMD materials. We also added the descriptions of the device structure combined with HAADF-TEM and EDX images.
5. Page 7, we added the descriptions of the reason for low electron-photon scattering in TMD materials combined with Raman spectra.
6. Page 8, we supplemented HAADF-TEM, EDX mapping, and Raman spectra characterizations in Figure 2. We reorganized Figure 2 and revised the figure captions. We also added a discussion of the performance comparison between commercial InGaAs and our WSe₂ diode.
7. Page 9, we added the descriptions of the evidence for avalanche breakdown and the photoelectric performance of our device.
8. Page 10, we added Figure 3. We also added a discussion of the effect of Schottky contact on the avalanche.
9. Page 11, we added a discussion of the band alignment and charge carrier transport in stepwise WSe₂ diodes.
10. Page 12, we reorganized Figure 4 and revised the figure captions.
11. Page 13, we added the calculation and analysis procedures for extracting the impact ionization rate.
12. Page 14, we supplemented the experimental data of *I-V* curves for the Schottky WSe₂ diode. We also reorganized Figure 5 and revised the figure captions.

Revised Supplementary Information:

1. Page 2, we reorganized the Supplementary Information and added the contents.
2. Page 3, we added the experimental data of thickness controllability for stepwise WSe₂ structures.
3. Page 4, we added the experimental data of carrier multiplication characteristic of the WSe₂ diode fabricated by selected area dry etching.
4. Page 5-11, we added a statistical analysis of twenty stepwise WSe₂ diodes with different thickness combinations. We also added detailed information including optical microscope images, AFM images, height profiles extracted from AFM results, and dark I - V curves of twenty devices.
5. Page 12, we reorganized Figure S9 and revised figure captions.
6. Page 14, we added a comparison of I - V curves between WSe₂ and InGaAs avalanche diodes.
7. Page 16, we added the experimental data of variable temperature I - V curves for two InGaAs avalanche diodes.
8. Page 17, we added a comparison of the temperature coefficient between WSe₂ and InGaAs avalanche diodes.
9. Page 18-19, we added the theoretical simulation results of band alignment for the stepwise WSe₂ diode.
10. Page 22, we added more photoelectronic properties of another WSe₂ avalanche diode.
11. Page 24-25, we added a comparison of performance metrics for different types of APDs.
12. Page 27, we added the experimental data of dark and photo-excited I - V curves for another three WSe₂ diodes.
13. Page 28, we added more experimental results of the BPV effect.
14. Page 29-30, we added the experimental data of response time.
15. Page 30-31, we added the experimental data of the stability characterizations.

REVIEWER COMMENTS

Reviewer #1 (Remarks to the Author):

Throughout this round of revision, the authors have thoroughly responded to the reviewers' comments raised in their point-by-point response letter and have significantly improved the quality of the manuscript with extra supporting experimental data. In principle, I would be inclined to re-consider my previous decision on this manuscript only if the following comments are properly addressed for establishing the validity of the key results that constitute the foundation of their work:

1. Temperature dependence of avalanche breakdown over a wider range:

a. $V_{\text{breakdown}}$ at lower temperature: The main message of the manuscript is the avalanche breakdown approaching the threshold limit set by the bandgap, E_g . To justify this interesting finding, the characterization of the avalanche breakdown, along with the extraction of $V_{\text{breakdown}}$, should be carried out in a wider range of temperatures (i.e. at least down to 77K, easily accessible with LN2). Their data shown in S11 and S13 should be extended to lower temperatures, ideally below 30K when we expect the el-ph scattering is expected to be minimal. From the device characteristics (ohmic contacts, as supported by SPCM in Fig.3 and other claims), it appears that there would be no significant issues for going down lower in temperature below 200 K.

b. Threshold limit of $V_{\text{breakdown}}$: If performed above, it would be reasonable to expect that the $V_{\text{breakdown}}$ would saturate to a certain value, which would be the threshold limit that the authors refer to, with a significantly less electron-phonon scattering. It would be not only a direct evidence for proving the above claim but their interpretation of the value of $V_{\text{breakdown}}$ being lower than E_g/e due to the built-in voltage.

c. Full characterisations at lower T: Their figures of merit for avalanche breakdown are likely to be better at lower temperatures, if their claim is accurate on the el-ph coupling. In addition to the $V_{\text{breakdown}}$ values, the avalanche gain, multiplication factor, α vs $1/E$, etc. should be measured and interpreted accordingly. One could expect even higher performance than the

2. Justification for the low el-ph scattering in WSe2 homojunction devices:

a. T-dep of mobility: In my opinion, it is not sufficient to claim the relatively low el-ph scattering in WSe2. As a suggestion, the above temperature dependence data can be correlated and interpreted with temperature-dependent mobility data (e.g. in FETs) for proving that they reach the threshold limit of $V_{\text{breakdown}}$ when there is minimal contribution of el-ph scattering.

b. Lower current at lower T: For the device shown in Fig. 2e, as well as Device 2 shown in Fig. S11b and S13b, the current seems to go down at lower T. The el-ph scattering would indicate the opposite. The authors could elaborate on this trend. One could argue that this is the finite contact resistance. If so, it would be important to explain why the $V_{\text{breakdown}}$ would go down at lower T despite the significantly larger voltage dissipation (as expected from the lower peak current shown in Fig. S11b) at the contact expected. Also see below.

c. Full range of data: The graphs shown in Fig. S11 have been presented differently in the V_{ds} range from the original graph (Fig. S3 in their originally submitted manuscript). The authors may be advised

against showing only a selective range of the data, especially when the full range of data may contain important information towards the mechanism of the avalanche breakdown (e.g. the forward current value being significantly lower at 280K than 300K (1 order of magnitude)).

d. hBN encapsulation: As pointed out by the reviewer #2's comment #1, placing the hBN flake should lower the $V_{\text{breakdown}}$ if the dominant bottleneck for avalanche breakdown was the el-ph coupling. I would be inclined to suggest reattempting the set of experiments since the authors have previously reported in papers a great level of expertise for dry-transfer techniques for preserving clean interfaces in 2D devices without the residue issues (Nature Electronics volume 4, pages399–404 (2021)).

* For your information- the effect of hBN is not just the surface recombination, as the authors write in the response letter, but the suppression of the out-of-plane phonon mode upon the top hBN encapsulation.

3. Built-in voltage at the homojunction

a. Further information for the doping characteristics used in TCAD: The authors present a nice TCAD calculation for calculating the built-in voltage of the band alignment at the junction. The doping profiles and parameters used for calculating Figure S14b is not so clear.

b. Experimental determination of built-in potential: The built-in potential at the junction should be relatively straightforward to extract from the diode curve, as well as SKPM. Complementing the simulation result with the experimental results will be very helpful since the interpretation of $V_{\text{breakdown}}$ outlined in the paper heavily depends on this value.

4. Novelty and message of the work

a. Interpretation and bigger picture: the authors should emphasize the advantages of their homojunction devices to numerous heterojunction devices reported in the paper, especially, in the lack of controllability and scalability in their fabrication method of such junction. Why should the WSe₂ homojunction device show a lower $V_{\text{breakdown}}$ that approaches the threshold limit even compared to InSe/BP heterojunction as reported in Nature Nanotech 2019? The low el-ph scattering is essential for achieving the quantum oscillations and ballistic transport that they observed in the paper, and it is difficult to understand that their device had not reached the threshold limit while the WSe₂ homojunction did.

b. Presentation of the figures: the current version of Fig. 1 is not very informative for clarifying the goal of their work. Perhaps, Fig.1 may be modified to contain 1) the schematics and characterization for their WSe₂ homojunction device, 2) the schematics for avalanche multiplications and 3) comparison with heterojunction devices.

c. Inclusion of heterojunction devices in Fig. 4c: Since the real 'competitors' for the reported device in the manuscript is the heterojunction diodes, the authors should include the heterojunction devices in the figure.

d. Necessity of BPV: I can understand why the authors would like to report that the BPV phenomenon is interesting, but this does not connect very well with the main theme of the manuscript, in my opinion. The authors could consider putting Fig. 5 and Fig. 6 in their supporting information.

Minor:

5. It would be helpful to present the extracted value for $V_{\text{breakdown}}$ at 200K on figure S13b for the

sake of completeness. It seems that the $V_{\text{breakdown}}$ at 200K for Device 2 might not fit very well with the linear fit shown in Figure S13b.

Reviewer #2 (Remarks to the Author):

It is good news that the manuscript has become stronger in many areas through the major revision process. However, while looking at the authors' responses, I would like to point out that concerns arose beyond curiosity about some aspects. Although the main message this manuscript is trying to convey may not change, the authors need to look more carefully at the discussion and conclusions in the additional data section.

1) For example, in the new devices using hBN, the author claims that the top BN layer is transferred to the WSe₂/BN layer where metal contacts are deposited. In this case, polymer residue still remains between the top hBN and WSe₂, and there is a high possibility that dirty impurities exist at the interface due to the thickness of the metal layer. The authors created a dirtier sample and took new measurements. My question is not whether the behavior of the avalanche is maintained with dirtier samples, but whether better quality samples can improve performance.

2) Authors claimed that they haven't observed a strong dependence of thickness combinations on the avalanche breakdown voltage. I think this conclusion is too hasty and dangerous. As already mentioned, the authors' samples were not made using the latest vdW assembly or metal contact method. The authors also did not identify monolayers (1 L, which will be the most dramatic junction gap change) in few-layer segments in 20 or more additional samples. In authors' reply as "we suggest that the thickness of the few-layer segment is 3-8 layers, while that of the thick part should ideally be more than 20 L", is the basis for this conclusion simple statistics? I doubt whether it can be concluded that the intrinsic limit of the avalanche threshold has been approached.

3) Looking at the comparison of WSe₂ and InGaAs avalanche diodes, InGaAs I-V is ultimately located in the upper left direction (threshold voltage goes negative with photo-generated current increase) compared to WSe₂. Even if high on-current is not welcome in avalanche photodetectors, can a photodetector with a working range of 10^{-10} ~ 10^{-14} A really be said to be good for application?

Reviewer #3 (Remarks to the Author):

In this resubmission, the Authors have addressed my concerns and comments raised previously. I feel that the overall quality of this work has been significantly improved – particularly to note that multiple new devices are fabricated and measured to address the comments from other reviewers. I also believe

that the manuscript has sufficiently addressed the comments raised by other reviewers. Many new “benchmark” studies and analysis were added in the revised manuscript, which makes the results and novelty very convincing to me. I thus recommend the manuscript to be accepted for publication in its current form.

Response to Reviewers' comments

We appreciate the referees for their valuable comments on the manuscript. We have carefully addressed all the comments and made revisions accordingly. In this response letter, the referees' comments are presented in black italic font, while our responses are in regular blue font. All major changes have been highlighted in blue in the main text and Supplementary Information.

Comments from the reviewers:

Reviewer #1 (Remarks to the Author):

Throughout this round of revision, the authors have thoroughly responded to the reviewers' comments raised in their point-by-point response letter and have significantly improved the quality of the manuscript with extra supporting experimental data. In principle, I would be inclined to re-consider my previous decision on this manuscript only if the following comments are properly addressed for establishing the validity of the key results that constitute the foundation of their work:

1. Temperature dependence of avalanche breakdown over a wider range:

a. $V_{\text{breakdown}}$ at lower temperature: The main message of the manuscript is the avalanche breakdown approaching the threshold limit set by the bandgap, Eg. To justify this interesting finding, the characterization of the avalanche breakdown, along with the extraction of $V_{\text{breakdown}}$, should be carried out in a wider range of temperatures (i.e. at least down to 77K, easily accessible with LN2). Their data shown in S11 and S13 should be extended to lower temperatures, ideally below 30K when we expect the el-ph scattering is expected to be minimal. From the device characteristics (ohmic contacts, as supported by SPCM in Fig.3 and other claims), it appears that there would be no significant issues for going down lower in temperature below 200 K.

b. Threshold limit of $V_{\text{breakdown}}$: If performed above, It would be reasonable to expect that the $V_{\text{breakdown}}$ would saturate to a certain value, which would be the threshold limit that the authors refer to, with a significantly less electron-phonon scattering. It would be not only a direct evidence for proving the above claim but their interpretation of the value of $V_{\text{breakdown}}$ being lower than E_g/e due to the built-in voltage.

Response: Thank you for your comments. Following your suggestions, we have

extended the operating temperature to below 200 K and reduced the temperature step to as low as 5 K. Through these efforts, we have observed a complete image of the breakdown voltage evolution properties. As depicted in Figure R1a, the breakdown voltage monotonously transitions from approximately 1.61 V at 292 K to around 0.70 V at 170 K. The temperature coefficient is then determined to be a positive value, which is consistent with our previous findings. Of particular interest is the significant decrease in the temperature coefficient as the temperature falls below 250 K (Figure R1c). This directly leads to a saturation tendency of the breakdown voltage. By fitting the curve with a double-exponential equation: $V_{br} = -A \times e^{\frac{T-T_0}{\alpha}} - B \times e^{\frac{T}{\beta}}$, we obtain a final limit value of -0.57 V ($0.35E_g/e$). Such a characteristic not only justifies our claim of the threshold-limit-avalanche but also explains why the breakdown voltage could be lower than E_g/e . This is because in the limit case, the internal built-in voltage also contributes to the avalanche process, thereby lowering the standard for external bias voltage, for example, $0.44E_g/e$.

Figure R1. Threshold limit of avalanche. Reverse-biased (a) and full-scale (b) I - V curves of the stepwise WSe₂ diode that was operated at decreasing temperatures. (c) Dependence of reverse breakdown voltage on the operating temperature.

We would like to present the full-scale I - V curves of the WSe₂ diode under different operating temperatures, as it provides more information on the temperature evolution property of the device. As illustrated in Figure R1b, the forward onset voltage (V_{on}) increases as the temperature decreases, which is in the opposite direction of the breakdown voltage. This phenomenon is associated with the broadened bandgap and lower carrier concentration at low temperatures. According to the equation $I_{forward} \propto e^{\frac{(qV-E_g(T))}{kT}}$, the forward current will not exponentially increase unless the external bias voltage is comparable to $E_g(T)/q$. Thus, the onset voltage will be proportional to the bandgap of the semiconductor, and at low temperatures, the threshold voltage will increase accordingly. Herein, $I_{forward}$ represents the forward current, q is the elementary charge, V is the external bias, k is the Boltzmann constant, and T is the temperature. We did not extend the temperature down to 77 K because the device does not function well at that temperature. Figure R2 shows that the WSe₂ diode typically becomes an insulator when the temperature drops below 170 K. Specifically, the forward current degrades rapidly from 34 pA to 20 fA @2 V as the temperature falls from 290 to 230 K (Figure R2a), and the reverse current degrades from 3 pA to 18 fA @-4 V in the temperature range of 215-170 K (Figure R2b). This degradation is reasonable because the charge carrier concentration decreases as the temperature drops, which hinders the forward conducting and reverse avalanche process. This effect is particularly significant in our WSe₂ diodes due to the extremely low background carrier concentration, as evidenced by the low dark current of 10 fA @ -0.1 V and 292 K. Therefore, our efforts primarily focus on the avalanche performance in the temperature range of 170-300 K.

Figure R2. I - V curves of the stepwise WSe₂ diode at decreasing temperatures.

c. Full characterisations at lower T : Their figures of merit for avalanche breakdown are likely to be better at lower temperatures, if their claim is accurate on the el-ph coupling. In addition to the $V_{\text{breakdown}}$ values, the avalanche gain, multiplication factor, α vs $1/E$, etc. should be measured and interpreted accordingly. One could expect even higher performance than the

Response: Thank you for your comments. Following your suggestions, we have extracted the figures of merit of the WSe₂ diodes, including the photo-excited I - V curve, multiplication factor, and impact ionization coefficient, at both room and low temperatures. Figure R3a illustrates the net photocurrent I - V curves of a WSe₂ diode at 300 and 260 K, respectively. It is evident that the avalanche breakdown voltage decreases from approximately 2.5 V at 300 K to around 1.7 V at 260 K, resulting in a significant increase in the device's photoresponse. Specifically, when we set the external bias to -2.5 V and adjust the operating temperature to 300 K, the device operates in the linear region, and the net photocurrent is measured to be 4 pA. In contrast, when the external bias remains constant and the temperature drops to 260 K, the device enters the avalanche region, causing the photocurrent to rise to 10 pA.

Figure R3. (a) Net photocurrent I - V curves of a WSe₂ diode at 300 and 260 K. (b) Extracted multiplication factor and (c) impact ionization coefficient of the device at 300 and 260 K.

Figure R3b compares the multiplication factor of the WSe₂ diode operated at 300 and 260 K. It is evident that the photogain/multiplication factor increases significantly as the temperature decreases from 300 to 260 K. In the linear region ($-1.7 \text{ V} < V_{ds} < -0.5 \text{ V}$), there is an approximately twofold increase in the value. Furthermore, in the avalanche region ($V_{ds} < -2.5 \text{ V}$), the gain is increased up to sevenfold. The data of impact ionization coefficients exhibit a similar trend (Figure R3c), with the impact ionization coefficient reaching $1.5 \times 10^4 \text{ cm}^{-1}$ at 260 K, approximately 1.2 times higher than that measured at 300 K. This behavior can be attributed to the variation in the intensity/strength of electron-phonon interaction at different temperatures. Generally, a decrease in temperature weakens the interaction between electrons and phonons, making it easier for charge carriers to acquire sufficient energy for the impact ionization process.

2. Justification for the low el-ph scattering in WSe2 homojunction devices:

a. T-dep of mobility: In my opinion, it is not sufficient to claim the relatively low el-ph scattering in WSe2. As a suggestion, the above temperature dependence data can be correlated and interpreted with temperature-dependent mobility data (e.g. in FETs) for proving that they reach the threshold limit of $V_{breakdown}$ when there is minimal contribution of el-ph scattering.

Response: Thank you for your questions.

(1) Weak electron-phonon interaction is one of the intrinsic properties of layered WSe₂, which has been extensively discussed in the literature. For example, recently, Fatemeh Barati et al. reported that the electron-phonon coupling in WSe₂/MoSe₂ heterostructures is intrinsically low, resulting in a slow cool-down of hot carriers and triggering interlayer charge carrier multiplication (Nat. Nanotechnol. 12, 1134-1139 2017). Ji-Hee Kim et al. reached a similar conclusion, demonstrating a carrier-multiplication efficiency of 99% in MoTe₂ and WSe₂ films (Nat. Commun. 10, 1-9 2019). The phonons in TMD materials can be divided into two groups: out-of-plane and in-plane modes. Generally, the out-of-plane modes (e.g., A_{1g} mode) exhibit stronger electron-phonon interactions compared to the in-plane modes (e.g., E_{1g} and E_{1u}). Furthermore, thinning the material can significantly weaken the contribution of the out-of-plane mode to electron-phonon scattering. This possibility is frequently verified through

Raman and inelastic-electron-tunneling spectroscopy experiments. In this study, we also present evidence from Raman spectra that supports this claim. Figure R4 displays the Raman spectra of WSe₂ with thicknesses of 4 L and 39 L, measured using a 532 nm laser line. The peak at 249.3 cm⁻¹ corresponds to the first-order Raman signal, which originates from the A_{1g} phonon mode. Additionally, there are peaks at 138.8, 257.5, 309.5, 360.6, 373.1, and 395.7 cm⁻¹, representing the second-order Raman signals associated with phonon combinations and overtones (ACS Nano 2014, 8, 9629). It is evident that the intensity of the second-order Raman signals decreases as the thickness of WSe₂ decreases from 39 L to 4 L. For example, the intensity of the 2LA(M) peak (at 257.5 cm⁻¹) decreases from 67% to 24% of the A_{1g} intensity. These results are consistent with previous reports on Raman spectroscopy of WSe₂ (Physical Review B 2013, 88, 195313; Physical Review B 2013, 87, 165409). This observation suggests that fewer phonons are active in WSe₂ as it approaches the monolayer limit, which may provide insight into the intrinsic weak electron-phonon interaction properties of transition metal dichalcogenide materials. We have also included a discussion on the weak electron-phonon interaction in the main text (Figure 2c).

Figure R4. Raman spectra of 4 L and 39 L WSe₂ measured with a 532 nm laser line. The spectra are normalized to the A_{1g} peak and vertically offset for clarity.

(2) Following your suggestions, we further fabricated additional uniform WSe₂ field-effect transistors. Considering that the depletion region primarily occurs in the few-layer WSe₂ (Figure 5b in the main text), charge carrier acceleration and impact ionization are likely to occur there. Therefore, our focus is to investigate the temperature dependence of mobility in few-layer WSe₂ devices. A typical dataset illustrating the temperature-dependent mobility is shown in Figure R5. The field-effect

mobility (μ) can be estimated by the equation $\mu = \frac{g_m L}{C_g V_{ds} W}$, where $g_m = \partial I_{ds} / \partial V_{gs}$ is the transconductance, L is the channel length, W is the channel width, C_g is the unit capacitance of the back gate, and V_{ds} is the applied voltage. By fitting the curve, it can be observed that the mobility starts to exhibit a saturation trend when the temperature drops below 200 K (Figure R5). This result provides evidence that the contribution of electron-phonon scattering can be disregarded and that the threshold voltage has reached its limit.

Figure R5. Field-effect mobility of a uniform WSe₂ transistor.

b. Lower current at lower T: For the device shown in Fig. 2e, as well as Device 2 shown in Fig. S11b and S13b, the current seems to go down at lower T. The el-ph scattering would indicate the opposite. The authors could elaborate on this trend. One could argue that this is the finite contact resistance. If so, it would be important to explain why the $V_{breakdown}$ would go down at lower T despite the significantly larger voltage dissipation (as expected from the lower peak current shown in Fig. S11b) at the contact expected. Also see below.

Response: Thank you for your questions. Indeed, there is a trend observed in the stepwise WSe₂ diode where the maximum avalanche current decreases as the temperature drops. Figure R6 illustrates an extreme case where the subthreshold swing of the reverse current degrades from 212 to approximately 2000 mV/dec as the temperature decreases from 300 to 240 K. As a result, the multiplication factor of the device is reduced by at least two orders. This phenomenon is attributed to the

competition between carrier multiplication and recombination processes, which are significantly influenced by the operating temperature. This effect has also been observed in low-temperature photoluminescence experiments, where a decrease in temperature leads to an increased rate of Auger recombination in multi-layer WSe_2 , resulting in fluorescence quenching (Figure R7, Adv. Optical Mater. 2019, 1901226; Nanoscale, 2018, 00, 1-3). This observation helps to explain the phenomenon observed in the stepwise WSe_2 diode. At low temperatures (e.g., 240 K), the Auger recombination rate, which is the reverse process of impact ionization, is high. Consequently, the charge carrier multiplication rate decreases, leading to a slow increase in reverse current (2000 mV/dec). On the other hand, the room temperature device benefits from a low Auger recombination rate, resulting in a fast increase in reverse current (212 mV/dec) within orders of magnitudes. Figure 6b clearly shows an intermediate state ($T=275$ K), where the two competing processes lead to a two-step rise in reverse current. The same trend can also be observed in Figure 2e (in the main text).

Figure R6. (a) and (b) Full-scale I - V curves of another WSe_2 diode that was operated at different temperatures. (c) Break-down and turn-on voltages derived from (a).

[REDACTED]

Figure R7. Fluorescence quenching in 7 L WSe₂ induced by temperature declines. (reproduced from Adv. Optical Mater. 2019, 1901226)

c. Full range of data: The graphs shown in Fig. S11 have been presented differently in the V_{ds} range from the original graph (Fig. S3 in their originally submitted manuscript). The authors may be advised against showing only a selective range of the data, especially when the full range of data may contain important information towards the mechanism of the avalanche breakdown (e.g. the forward current value being significantly lower at 280K than 300K (1 order of magnitude)).

Response: Thanks for your suggestion. We have provided the full range of I - V curves of three WSe₂ diodes under different temperatures (Figures R1, R2, and R6; Figure 3 in the main text). Obviously, the reverse breakdown voltage shows a positive temperature coefficient due to the ever-weakened electron-phonon interactions. This is clear evidence of avalanche breakdown. By contrast, the forward turn-on voltage shows a negative temperature coefficient due to the decreased/increased charge-carrier-concentration/bandgap.

d. hBN encapsulation: As pointed out by the reviewer #2's comment #1, placing the hBN flake should lower the $V_{breakdown}$ if the dominant bottleneck for avalanche

breakdown was the el-ph coupling. I would be inclined to suggest reattempting the set of experiments since the authors have previously reported in papers a great level of expertise for dry-transfer techniques for preserving clean interfaces in 2D devices without the residue issues (*Nature Electronics* volume 4, pages399–404 (2021)).

* For your information- the effect of hBN is not just the surface recombination, as the authors write in the response letter, but the suppression of the out-of-plane phonon mode upon the top hBN encapsulation.

Response: Thank you for your questions. We have fabricated additional WSe₂/BN diodes and performed TEM measurements to verify the achievement of a clean interface after dry-transfer processes (Figure R8). In this case, we conducted a statistical analysis of the breakdown voltage data for 11 WSe₂/hBN diodes in comparison to 25 pristine WSe₂ diodes. As shown in Figure R9, the threshold voltage of the WSe₂/hBN and pristine WSe₂ diodes ranges from -1.2 to -1.8 V (indicated by the red bars) and from -1.4 to -5.4 V (indicated by the blue bar), respectively. Clearly, WSe₂/hBN diodes exhibit lower minimum threshold voltages (-1.2 V, devices #5 and #6 in Figure R10) and a more concentrated distribution compared to pristine WSe₂. For a more detailed analysis, we divided the threshold voltage into three ranges: $|V_{br}| < 1.5$ V, 1.5 V $\leq |V_{br}| < 2$ V, and $|V_{br}| \geq 2$ V. Among them, the proportions of WSe₂/hBN and pristine WSe₂ diodes in the three threshold voltage ranges are 36%, 64%, 0%, and 20%, 40%, 40%, respectively. This further indicates that the overall performance of WSe₂ diodes is indeed improved when a hBN layer is used as the substrate, due to the reduced scattering processes.

Figure R8. High-resolution TEM images near the WSe₂/hBN interface.

Figure R9. Comparison of threshold voltage distribution for WSe₂/hBN (left panel) and pristine WSe₂ diodes (right panel). The red and Blue bars indicate the threshold voltage distribution ranges of 11 WSe₂/hBN and 25 WSe₂ diodes, respectively.

Figure R10. I - V curves of Numbers #5, #6, and #7 WSe₂ diodes encapsulated with BN in the dark and under illumination at 300 K. Inset: optical microscope images of the devices.

3. Built-in voltage at the homojunction

a. Further information for the doping characteristics used in TCAD: The authors present a nice TCAD calculation for calculating the built-in voltage of the band alignment at the junction. The doping profiles and parameters used for calculating Figure S14b is not so clear.

Response: Thank you for your questions. The theoretical model was established using a commercial software package, Sentaurus-TCAD. It takes into account the stepwise geometry of the WSe₂ device, with two distinct segments of thickness 5.6 (8 L) and 20.3 nm (29 L). The electron affinity and bandgaps for the few-layer and multi-layer parts are 3.7 and 1.6 eV, and 4.0 and 1.2 eV, respectively (Opt. Quantum Electron., 2020, 52, 1-14; Appl. Phys. Lett. 2013, 102, 012111).

As shown in Figure R11, the built-in potential arises from two factors: the band offset and the Fermi-level drop, induced by different doping polarities and concentrations in the counterpart segments. According to the simulation, the band offset alone results in an internal potential of 0.41 V ($\frac{1.6 \text{ eV} - 1.2 \text{ eV}}{e} = 0.4 \text{ V}$, Figure R11a), while the Fermi-level drop increases it to 1.18 V and even higher (Figure R11). In the former case, the doping concentration is unified for the few-layer and multi-layer WSe₂, $n = 1 \times 10^{15} \text{ cm}^{-3}$. In the latter case, the doping concentration is $n = 1 \times 10^{16} \text{ cm}^{-3}$ and $-1.5 \times 10^{16} \text{ cm}^{-3}$, respectively.

Figure R11. Calculated energy-band structure of WSe₂ diode at zero bias voltage. (a) without and (b) with considering the discrepancy of doping characteristics between the few- and multi-layer segments. n_{2D} and n_{3D} represent the two- and three-dimensional carrier concentration, that are counted in the form of cm^{-2} and cm^{-3} .

b. Experimental determination of built-in potential: The built-in potential at the junction should be relatively straightforward to extract from the diode curve, as well as SKPM. Complementing the simulation result with the experimental results will be very helpful since the interpretation of $V_{\text{breakdown}}$ outlined in the paper heavily depends on this value.

Response: Thank you for your questions. To address this issue, we conducted scanning Kelvin probe microscopy (SKPM) experiments on the stepwise WSe₂ structure. As depicted in Figure R12, the tested WSe₂ flake comprises various layer thicknesses, including 3 L, 4 L, 5 L, 6 L, 8 L, and 10 L. Consequently, the surface potential transitions from 0.076 V to 0.185, 0.53, 0.618, 0.991, and finally 1.173 V (Figure R12). This indicates that the Fermi-level difference or built-in potential in the stepwise WSe₂ structure can reach up to 1.1 eV, consistent with theoretical calculations (Figure R11, 1.18 eV). Based on this, the minimum voltage required for the avalanche would be reduced to $(\frac{E_g=1.6 \text{ eV}}{e} - 1.1 \text{ V} = 0.5 \text{ V})$. This explains why the breakdown voltage can further decrease as the temperature drops.

Figure R12. SKPM experiment on the stepwise WSe₂ structure. The left and middle panels are the topography and surface potential images of the structure, respectively. The right panel is the potential profile derived from the middle image.

4. Novelty and message of the work

a. Interpretation and bigger picture: the authors should emphasize the advantages of their homojunction devices to numerous heterojunction devices reported in the paper, especially, in the lack of controllability and scalability in their fabrication method of such junction. Why should the WSe₂ homojunction device show a lower $V_{\text{breakdown}}$ that approaches the threshold limit even compared to InSe/BP heterojunction as reported in Nature Nanotech. 2019? The low el-ph scattering is essential for achieving the quantum oscillations and ballistic transport that they observed in the paper, and it is difficult to understand that their device had not reached the threshold limit while the WSe₂ homojunction did.

Response: Thank you for your questions.

(1) Controllability in the device fabrication

In addition to the standard mechanical exfoliation process, there is an alternative method for shaping vdW materials, namely the selective area dry-etching process. In this study, we also present the device fabricated using this technique.

Figure R13. Carrier multiplication characteristic of the WSe₂ diode fabricated by RIE etching. (a) Optical microscope and AFM image of the stepwise WSe₂ diode by RIE method. (b) Height profile showing the thickness of few-layer and multilayer WSe₂. (c) Dark I - V curve of the WSe₂ diode at 300 K.

Figure R13a illustrates the procedure for fabricating the device, which allows for full control over the morphology of van der Waals materials. First, we utilize state-of-the-art lithography to expose half of the multilayer WSe₂ flake. This is followed by a standard reactive ion dry etching (RIE) process, using plasma species such as Ar and CF₄ to thin the exposed WSe₂ flake to 3 L. Subsequently, Pt/Au electrodes are deposited on both sides (34 L and 3 L) to ensure electrical contact.

We then proceed to characterize the avalanche performance of the RIE-etched WSe₂ diode. Figure R13c shows the I - V curve of the prepared WSe₂ diode, demonstrating significant avalanche breakdown at a low bias voltage (-6 V). This further confirms that the low threshold avalanche is an intrinsic property of the stepwise WSe₂ diode.

It is worth noting that the breakdown voltage of the etched diode is somewhat higher than that of the mechanically exfoliated device. This can be attributed to the crystal damage induced by the RIE process, which may increase scattering events and thus reduce the efficiency of charge carrier acceleration. Recently, there have been reports

of a van der Waals peeling technique that allows for layer-by-layer thinning of materials (Nat. Electron. 2023. <https://doi.org/10.1038/s41928-023-01087-8>). This approach avoids lattice damage during the thinning process and has the potential to improve avalanche performance.

(2) Discrepancy in the breakdown mechanism between stepwise WSe₂ and InSe/BP diode

As documented in the textbook, avalanche breakdown is characterized by a positive temperature coefficient, where the breakdown voltage decreases as the temperature decreases. The underlying physics can be described as follows: as the temperature decreases, electrons experience fewer scattering events with phonons (due to the freezing of lattice vibrations), allowing them to easily acquire excess energy (E_g) and initiate the impact ionization process. Consequently, the external electric voltage/power required for avalanche breakdown decreases continuously.

In this study, we fabricated diodes using both InGaAs and stepwise WSe₂ materials. As depicted in Figure R14, both diodes exhibit a clear positive temperature coefficient, confirming the dominance of the avalanche process in the breakdown current. However, in the InSe/BP diode (Nature Nanotechnology 2019, 14, 217), the temperature coefficient is observed to be negative. As mentioned in this literature, “A comprehensive understanding of this phenomenon requires further theoretical analysis and sophisticated transport measurements, which remain open questions at this stage.” Therefore, at present, we are unable to provide a physical explanation for why the breakdown voltage of the InSe/BP diode fails to reach the threshold limit. This issue is beyond the scope of our current work.

Figure R14. Dependence of breakdown voltage on the operating temperature in InGaAs, stepwise WSe₂, and InSe/BP avalanche diodes. The scatter dots are the experimental data while the solid lines are added for eye guidance. A#, B#, and C# represent three different stepwise WSe₂ diodes, which are shown in Figures R1, R2, and R6. The data of the InSe/BP avalanche diode is reproduced from the literature Nature Nanotechnology 2019, 14, 217.

(3) Superiority of WSe₂ homojunction in achieving low threshold avalanche.

The WSe₂ homojunction possesses two intriguing properties: weak electron-phonon interaction and strong internal electric field. Together, these aspects reduce the breakdown voltage to the threshold limit.

Weak electron-phonon interaction. This is evidenced by the Raman experiments. Figure R15 presents the Raman spectra of 4 L and 39 L WSe₂, measured using a 532 nm laser line. The peak at 249.3 cm⁻¹ represents the first-order Raman signal, attributed to the A_{1g} phonon mode. Additional peaks at 138.8, 257.5, 309.5, 360.6, 373.1, and 395.7 cm⁻¹ correspond to the second-order Raman signals, associated with combination and overtones of phonons. Notably, the intensity of the second-order Raman signals decreases as the thickness of WSe₂ scales from 39 L to 4 L. For instance, the intensity of 2LA(M) (peaked at 257.5 cm⁻¹) declines from 67% to 24% of the A_{1g} intensity. This result is consistent with previous reports and suggests that fewer phonons are active in WSe₂ as its thickness approaches the monolayer limit. Considering that the few-layer WSe₂ is the multiplication region, these findings may provide insights into the intrinsic low breakdown voltage.

Figure R15. Raman spectra of 4 L and 39 L WSe₂ measured with a 532 nm laser line. The spectra are normalized to the A_{1g} peak and vertically offset for clarity.

Enhanced internal electric field. The stepwise morphology results in an atomically abrupt homojunction, in which the few-/multi-layer segment acts as a wide-/narrow-

gap semiconductor, respectively. Moreover, this device architecture accumulates more electric field energy at the depletion region, thereby triggering the charge carrier avalanche process earlier. Numerical simulations reveal that the peak electric field of the vdW junction is approximately 4 times higher than that of the traditional counterpart (bottom panel of Figure R16). Consequently, the external bias voltage required for the avalanche would be nearly 16 times lower, reducing from -24.5 to -1.6 V.

Figure R16. Top panel: the architecture and electric field distribution of a stepwise TMD and bulk junction. Bottom panel: the dependence of peak electric field on the reverse biased voltage for TMD and bulk junction.

b. Presentaiotn of the figures: the current version of Fig. 1 is not very informative for clarifying the goal of their work. Perhaps, Fig.1 may be modified to contain 1)the schematics and characterization for their WSe₂ homojunction device, 2) the schematics for avalanche multiplications and 3)comparison with heterojunction devices.

Response: Thank you for your comments. In the current version of the manuscript, Figure 1 demonstrates the superiority of the stepwise WSe₂ homojunction in achieving charge carrier avalanche theoretically, while Figure 2 presents experimental evidence for this phenomenon.

In Figure 1, the top panel schematically illustrates the electron-phonon scattering events and their occurrence rates in bulk and van der Waals materials. Generally, intense electron-phonon scatterings occur in bulk materials, resulting in significant energy loss during the charge carrier acceleration process and thus delaying the impact ionization process. Conversely, in van der Waals materials, the electron-phonon interactions

weaken as the layer thickness shrinks to the monolayer limit. This leads to a lower loss of energy, initiating the impact ionization at an earlier stage. The bottom panel of Figure 1 displays the electric field distributions in bulk and van der Waals junctions. Numerical simulations indicate that the peak electric field of the vdW junction is approximately four times higher than that of the traditional counterpart. Consequently, the external bias voltage required for avalanche would be nearly 16 times lower, reducing from -24.5 to -1.6 V. This relationship is determined by the equation: $E \propto \sqrt{V_{ex}}$, where E and V_{ex} represent the electric field and external bias voltage, respectively.

Figure 2 presents the experimental results, which include “1) the schematics and characterization for the WSe₂ homojunction device” and “2) the schematics for avalanche multiplications”. These results confirm the presence of charge carrier avalanche in the stepwise WSe₂ diode, with a breakdown voltage of approximately 1.6 V, nearly two orders of magnitude lower than that of the bulk device.

Considering the aforementioned factors, we maintain the arrangement of Figures 1 and 2. Figure 1 theoretically demonstrates the superiority of the stepwise WSe₂ homojunction in achieving a low threshold avalanche, while Figure 2 provides experimental evidence for this.

c. Inclusion of heterojunction devices in Fig. 4c: Since the real ‘competitors’ for the reported device in the manuscript is the heterojunction diodes, the authors should include the heterojunction devices in the figure.

Response: Thank you for your comment. We have included the results of the van der Waals heterojunction for comparison. Figure R17 and Table R1 show the breakdown voltage of bulk material, van der Waals material, and our stepwise WSe₂ structure at room temperature. It is evident that the threshold voltage in the bulk material is as high as 150 V. This necessitates the design of special driving and signal-processing circuits, which significantly increases the cost and limits the applications. Recently, researchers have explored avalanche devices based on uniform MoS₂, WSe₂, MoS₂/WSe₂, and WS₂/WSe₂ heterostructures, inspired by the fascinating properties of vdW materials. These efforts have reduced the avalanche voltage to around 10 V at 300 K, representing a significant advancement. However, the breakdown voltage still exceeds the range of traditional digital circuits, which operate within a ± 5 V voltage range. In this study, we

demonstrate that the stepwise WSe₂ architecture can further decrease the threshold voltage to 1.6 V. This achievement approaches the classical limit of avalanching, where the external energy required for avalanching is equivalent to the energy gap of the multiplication region, $V_{ex} \times q \approx E_g = 1.6$ eV (for bulk avalanche diodes, the threshold energy must reach $22E_g$ or higher).

Figure R17. Summary on the room temperature breakdown voltage of bulk and vdW avalanche diodes. The middle four panels: reproduced from ACS Nano 2018, 12, 7, 71096, Nano Res. 2023, 16, 3422, and Nano Lett. 2022, 22, 9516.

More details of these avalanche photodetectors (APDs) can be found in Table 1. Typically, APDs based on conventional bulk materials exhibit good performance at room temperature but often require a high breakdown voltage (greater than 25 V). In addition, all bulk materials-based APDs display a positive temperature coefficient, which is a characteristic feature of the avalanche multiplication process. In comparison, APDs based on two-dimensional materials have a relatively low breakdown voltage but typically operate at low-temperature conditions. Our study proposes a WSe₂ stepwise homojunction diode with a low avalanche threshold voltage of approximately 1.6 V, a low dark current of about 10 fA, and a large avalanche gain of approximately 200 at room temperature. Importantly, the proposed WSe₂ diodes demonstrate a positive temperature coefficient, similar to that of traditional bulk APDs.

Table R1. Comparison of performance metrics for different types of APDs

APD structure	Operating temperature	Temperature coefficient	Breakdown voltage	Gain	Dark current	Ref.	
Bulk material	Si PIN junction	300 K	Positive	-150 V	~100 @-150 V	~50 pA	1
	InGaAs PIN junction	300 K	Positive	-55 V	~30 @-55 V	~40 nA	2
	Si-Ge PIN junction	300 K	Positive	-25 V	~30 @-25 V	~100 pA	3
	GaN PIN junction	300 K	Positive	-92 V	~300 @-92 V	~100 pA	4
	Ge PIN junction	300 K	Positive	-30 V	~100 @-30 V	~300 nA	5
	AlGaIn Schottky junction	/	/	-50 V	1560 @-68 V	~10 pA	6
Two-dimensional material	WSe ₂ /WS ₂ PN heterojunction	/	/	-8.5 V	~300 @-16.5 V	~100 pA	7
	MoS ₂ /WSe ₂ PN heterojunction	300 K	/	-6.5 V	~5 @-10 V	~100 pA	8
	MoS ₂ PN homojunction	100 K	/	-4.5 V	~100 @-10 V	~1 pA	9
	MoS ₂ Uniform n-doping	300 K	positive	-40 V	/	~1 μA	10
	BP/InSe PN heterojunction	80 K	negative	-4.8 V	~100 @-2 V	~100 pA	11
	WSe₂ Stepwise homojunction	300 K	Positive	-1.6 V	~200 @-2 V	~10 fA	This work

[1] <https://www.hamamatsu.com.cn/cn/zh-cn/product/optical-sensors/apd/si-apd/S12023-02.html>

[2] <https://www.hamamatsu.com.cn/cn/zh-cn/product/optical-sensors/apd/ingaas-apd/G8931-04.html>

[3] Kang Y, Liu H, Morse M, et al. Monolithic germanium/silicon avalanche photodiodes with 340 GHz gain–bandwidth product. Nature Photonics, 2009, 3, 59–63.

[4] Verghese S, McIntosh K A, Molnar R J, et al. GaN avalanche photodiodes operating in linear-gain mode and Geiger mode. IEEE Transactions on Electron Devices, 2001, 48(3): 502-511.

[5] <https://www.ushio.com/product/pd-ld-germanium-avalanche-photodiode/>.

[6] Tut T, Gokkavas M, Inal A, et al. Al_xGa_{1-x}N-based avalanche photodiodes with high reproducible avalanche gain. Applied Physics Letters, 2007, 90, 163506.

[7] Meng L, Zhang N, Yang M, et al. Low-voltage and high-gain WSe₂ avalanche phototransistor with an out-of-plane WSe₂/WS₂ heterojunction. Nano Research, 2023,

16, 3422-3428.

[8] Son B, Wang Y, Luo M, et al. Efficient Avalanche Photodiodes with a WSe₂/MoS₂ Heterostructure via Two-Photon Absorption. *Nano letters*, 2022, 22, 9516-9522.

[9] Xia H, Luo M, Wang W, et al. Pristine PN junction toward atomic layer devices. *Light: Science & Applications*, 2022, 11, 170.

[10] Pak, Jinsu, et al. Two-dimensional thickness-dependent avalanche breakdown phenomena in MoS₂ field-effect transistors under high electric fields. *ACS Nano* 2018, 12, 7109.

[11] Gao A, Lai J, Wang Y, et al. Observation of ballistic avalanche phenomena in nanoscale vertical InSe/BP heterostructures. *Nature Nanotechnology*, 2019, 14(3): 217-222.

d. Necessity of BPV: I can understand why the authors would like to report that the BPV phenomenon is interesting, but this does not connect very well with the main theme of the manuscript, in my opinion. The authors could consider putting Fig. 5 and Fig. 6 in their supporting information.

Response: Thanks for your comments. According to your suggestions, we have removed the content of the BPV phenomenon (Figure 5 and Figure 6) from the main text.

Minor:

5. It would be helpful to present the extracted value for $V_{breakdown}$ at 200K on figure S13b for the sake of completeness. It seems that the $V_{breakdown}$ at 200K for Device 2 might not fit very well with the linear fit shown in Figure S13b.

Response: Thank you for your advice. Figure R14 shows the dependence of breakdown voltage on the operating temperature in 3 different devices, A#, B#, and C# devices. The full-scale $I-V$ curves of the three devices are shown in Figures R1, R2, and R6 (Figure 3 in the main text). Indeed, they are not in a linear fit but follow a saturation relationship. Such characters further confirm that the stepwise WSe₂ didoes reach the threshold limit of avalanche limit. More detailed discussion could be found in the discussion @Q1 or the main text. We have also made modifications to these contents in the Supplementary Information (Figure S14).

Reviewer #2 (Remarks to the Author):

It is good news that the manuscript has become stronger in many areas through the major revision process. However, while looking at the authors' responses, I would like to point out that concerns arose beyond curiosity about some aspects. Although the main message this manuscript is trying to convey may not change, the authors need to look more carefully at the discussion and conclusions in the additional data section.

1) For example, in the new devices using hBN, the author claims that the top BN layer is transferred to the WSe₂/BN layer where metal contacts are deposited. In this case, polymer residue still remains between the top hBN and WSe₂, and there is a high possibility that dirty impurities exist at the interface due to the thickness of the metal layer. The authors created a dirtier sample and took new measurements. My question is not whether the behavior of the avalanche is maintained with dirtier samples, but whether better quality samples can improve performance.

Response: Thank you for your valuable suggestions. We acknowledge the potential presence of residues at the mental interface, which may have a negative impact on the performance of the device. Therefore, following your advice, we initially transferred the hBN layer onto the Si/SiO₂ substrate and then transferred WSe₂ onto the surface of hBN to create the WSe₂/hBN diodes. Additionally, we conducted high-resolution TEM tests to confirm the formation of a high-quality WSe₂/BN interface after the transfer processes. As illustrated in Figure R1, the TEM images reveal no noticeable impurities or defects near the WSe₂/hBN interface, enabling us to investigate the impact of the hBN layer on device performance more effectively.

Figure R1. High-resolution TEM images near the WSe₂/hBN interface.

We fabricated additional WSe₂/hBN diodes and conducted a statistical analysis of the threshold voltage data for 11 WSe₂/hBN diodes in comparison to 25 pristine WSe₂ diodes. The results are presented in Figure R2. Among these devices, the threshold voltage of the WSe₂/hBN and pristine WSe₂ diodes range from -1.2 to -1.8 V (indicated by the red bars) and from -1.4 to -5.4 V (indicated by the blue bar), respectively. Clearly, WSe₂/hBN diodes exhibit lower minimum threshold voltages (-1.2 V, devices #5 and #6 in Figure R3) and a more concentrated distribution compared to pristine WSe₂. For a more detailed analysis, we divided the threshold voltage into three ranges: $|V_{br}| < 1.5$ V, $1.5 \text{ V} \leq |V_{br}| < 2$ V, and $|V_{br}| \geq 2$ V. Among them, the proportions of WSe₂/hBN and pristine WSe₂ diodes in the three threshold voltage ranges are 36%, 64%, 0%, and 20%, 40%, 40%, respectively. This further indicates that the overall performance of WSe₂ diodes is indeed improved when a hBN layer is used as the substrate, due to the reduced scattering processes. The detailed experimental data, including photo-excited I - V curves and threshold voltage distribution for 11 WSe₂/hBN and 25 pristine WSe₂ diodes, are shown in Figure R3-5 and Table R1.

Figure R2. Comparison of threshold voltage distribution for WSe₂/hBN (left panel) and pristine WSe₂ diodes (right panel). The red and Blue bars indicate the threshold voltage distribution ranges of 11 WSe₂/hBN and 25 WSe₂ diodes, respectively.

Figure R3. I - V curves of Numbers #5, #6, and #7 WSe₂ diodes encapsulated with BN in the dark and under illumination at 300 K. Inset: optical microscope images of the devices.

Figure R4. I - V curves of Numbers #8 and #9 WSe₂ diodes encapsulated with hBN in the dark and under illumination at 300 K. Inset: optical microscope images of the devices.

Figure R5. I - V curves of Numbers #10 and #11 WSe₂ diodes encapsulated with BN in the dark and under illumination at 300 K. Inset: optical microscope images of the devices.

Table R1. Summary of breakdown voltage for WSe₂/hBN and pristine WSe₂ devices.

Breakdown voltage Range	Counts of the device		Device Number(#)	
	WSe ₂ /BN	Pristine WSe ₂	WSe ₂ /BN	Pristine WSe ₂
$ V_{br} < 1.5$ V	4	5	#1: -1.4 V; #5, #6: -1.2 V; #7: -1.3 V	#1, #4, #7, #10, #14: -1.4 V
1.5 V $\leq V_{br} < 2$ V	7	10	#2, #3, #8, #9: -1.5 V; #4, #10: -1.7 V; #11: -1.8 V	#3, #8: 1.9 V; #6: 1.7 V; #5, #15: -1.6 V; #9, #11, #13, #16: -1.5 V; #12: -1.8 V
$ V_{br} \geq 2$ V	0	10	/	#2: -2.1 V; #17, #25: -3 V; #18: -2.2 V; #19, #21: -5 V; #20: -5.4 V; #22: -4.8 V; #23: -3.5 V; #24: -4.3 V

2) Authors claimed that they haven't observed a strong dependence of thickness combinations on the avalanche breakdown voltage. I think this conclusion is too hasty and dangerous. As already mentioned, the authors' samples were not made using the latest vdW assembly or metal contact method. The authors also did not identify monolayers (1 L, which will the most dramatic junction gap change) in few-layer segments in 20 or more additional samples. In authors' reply as "we suggest that the thickness of the few-layer segment is 3-8 layers, while that of the thick part should ideally be more than 20 L", is the basis for this conclusion simple statistics? I doubt whether it can be concluded that the intrinsic limit of the avalanche threshold has been approached.

Response: Thank you for your valuable comments. In the previously submitted response letter, we summarized the breakdown voltages and thickness combinations of 20 WSe₂ devices (Figure R6, where each scatter dot represents an individual device). It is evident that there is a relatively discrete distribution of breakdown voltages among these devices, which leads us to the conclusion that "there is no strong dependence of thickness combinations on the avalanche breakdown voltage".

Figure R6. Summary on the breakdown voltage as well as the thickness combinations of 20 WSe₂ devices.

Figure R7. Statistical analysis on the dependence of breakdown voltage on the few-layer WSe₂ thickness.

In this section, we will follow your suggestions and conduct a rigorous statistical analysis. Considering that the depletion region primarily occurs in the few-layer WSe₂ (Figure 5b in the main text), charge carrier acceleration and impact ionization are likely to occur there. Therefore, our focus is on the impact of the few-layer thickness on the breakdown voltage. As shown in Figure R7, we divided 25 devices (including 5 recently fabricated ones, with optical microscope and AFM images as well as I - V curves shown in Figure R8-R9) into two groups: $1 \text{ V} < |V_{\text{br}}| < 2.5 \text{ V}$ and $2.5 \text{ V} < |V_{\text{br}}| < 7 \text{ V}$. Within each group, the devices are further subdivided into 4 sections based on the few-layer thickness: $\leq 3 \text{ L}$, $4\text{-}6 \text{ L}$, $7\text{-}9 \text{ L}$, and $\geq 10 \text{ L}$. This approach will enable us to easily establish the relationship between the breakdown voltage and the thickness of WSe₂. As shown in Figure R7, high breakdown voltage (2.5-7 V) is frequently observed in thin devices, with the thickness of the few-layer WSe₂ being approximately $\sim 4 \text{ L}$. In contrast, a low breakdown voltage (1-2.5 V) is closely associated with thick devices, where the thickness of the few-layer WSe₂ is approximately $\sim 7 \text{ L}$. This is reasonable because the bandgap is inversely proportional to the thickness of TMD materials. Therefore, for avalanche breakdown, thinner/thicker materials (with larger/smaller bandgaps) will require a higher/lower external bias voltage. Detailed information on thickness

combinations and breakdown voltage for 25 WSe₂ devices is listed in Table R2. Additionally, it needs to be clarified that the avalanche process is influenced by several inherent factors, including doping concentration, the width of the avalanche region, bandgap, and even device shape (for low-dimensional materials). Therefore, fluctuations in the breakdown voltage of avalanche diodes are commonly observed. For instance, there is a variation of up to 1 V in the breakdown voltage among 10 commercial InGaAs avalanche devices, despite their identical design, as illustrated in Figure R10. Therefore, based on the above discussion and statistical analysis results, we can conclude that thickness combinations are crucial for the avalanche breakdown voltage. However, to be more rigorous, we believe it is important to note that thickness combinations are not the only factor to consider for achieving low-threshold avalanche devices.

Table R2. Summary of the stepwise WSe₂ diodes with different thickness combinations.

Device Number (#)	Few-layer WSe ₂ (layer)	Multilayer WSe ₂ (layer)	Breakdown voltage (V)
1	~6	~29	-1.4
2	~5	~36	-2.1
3	~3	~17	-1.9
4	~6	~28	-1.4
5	~3	~27	-1.6
6	~8	~25	-1.7
7	~7	~33	-1.4
8	~5	~75	-1.9
9	~7	~19	-1.5
10	~3	~55	-1.4
11	~8	~29	-1.5
12	~7	~22	-1.8
13	~7	~38	-1.5
14	~6	~53	-1.4
15	~7	~45	-1.6

16	~5	~27	-1.5
17	~13	~22	-3.0
18	~10	~53	-2.2
19	~3	~13	-5.0
20	~5	~54	-5.4
21	~4	~46	-5.0
22	~7	~36	-4.8
23	~4	~45	-3.5
24	~3	~16	-4.3
25	~6	~40	-3.0

a) The thickness of the monolayer WSe₂ is about 0.7 nm.

Figure R8. Optical microscope images, AFM images, height profiles extracted from AFM results, and dark I - V curves of Numbers #12, #13, and #14 stepwise WSe_2 diodes.

Figure R9. Optical microscope images, AFM images, height profiles extracted from AFM results, and dark I - V curves of Numbers #15 and #16 stepwise WSe_2 diodes.

Figure R10. Fluctuations of breakdown voltage among 10 commercial InGaAs avalanche devices

3) Looking at the comparison of WSe₂ and InGaAs avalanche diodes, InGaAs I-V is ultimately located in the upper left direction (threshold voltage goes negative with photo-generated current increase) compared to WSe₂. Even if high on-current is not welcome in avalanche photodetectors, can a photodetector with a working range of 10^{-10} ~ 10^{-14} A really be said to be good for application?

Response: Thank you for your comments. We would like to demonstrate the case of a commercial InGaAs PIN diode, which will help address this question. As shown in Figure R11, the dark current of the InGaAs PIN diode is as low as 10 fA, similar to our WSe₂ diode. In practical applications, this is considered an advantage rather than a drawback for light signal detection. This is because, under such conditions, the photo-generated current easily surpasses the background current and can be detected. Table R3 summarizes the dark current and minimum detectable photons of a photodetector with an external quantum efficiency set at 100%. It can be observed that each percent decrease in dark current corresponds to an increase in sensitivity of the same magnitude. For this reason, most commercial InGaAs PIN diodes operate in a low noise region, such as $V = -100$ mV and $I = 17$ fA (Infrared Technology and Applications XXXVIII. SPIE, 2012, 8353: 88-95).

Figure R11. Dark current I - V curves of commercial InGaAs PIN diode and our WSe₂ diode. The data of InGaAs PIN diode is reproduced from Infrared Technology and Applications XXXVIII. SPIE, 2012, 8353, 88-95.

Table R3. Dependence of minimum detectable photons on the dark current of photodetectors.

Dark current (A)	Minimum detectable photons
1×10^{-6}	6.3×10^{12}
1×10^{-10}	6.3×10^8
1×10^{-14}	6.3×10^4

In practical applications, the low-noise photodetector will be combined with a readout integrated circuit (ROIC) to achieve the detection capability. If the noise is expressed in electrons, the noise level of an operating InGaAs PIN diode ($V=-100$ mV) is approximately $80 e^-$, while the noise level of the ROIC is around $100 e^-$ (Infrared Technology and Applications XXXVIII. SPIE, 2012, 8353: 88-95; Infrared Technology and Applications XXXVIII. SPIE, 2012, 8353, 116-122; Sensors, Cameras, and Systems for Industrial and Scientific Applications XIII. SPIE, 2012, 8298, 101-108). This indicates that these ROICs are in the same low-noise range, enabling them to easily read out the photo-induced signal from the detectors.

Reviewer #3 (Remarks to the Author):

In this resubmission, the Authors have addressed my concerns and comments raised previously. I feel that the overall quality of this work has been significantly improved – particularly to note that multiple new devices are fabricated and measured to address the comments from other reviewers. I also believe that the manuscript has sufficiently addressed the comments raised by other reviewers. Many new “benchmark” studies and analysis were added in the revised manuscript, which makes the results and novelty very convincing to me. I thus recommend the manuscript to be accepted for publication in its current form.

Response: Thank you for your valuable input. I sincerely appreciate the efforts made by the Reviewers to provide valuable comments and suggestions previously. It is evident that these suggestions have significantly improved the overall quality of our work. Thanks again for your valuable time and efforts.

REVIEWERS' COMMENTS

Reviewer #1 (Remarks to the Author):

In the revised manuscript, the authors have fully revised their manuscript and performed various extra measurements to support their claims on the near-threshold avalanche breakdown behavior in WSe₂ homojunction devices. Although I feel that the study lacks fundamental accounts for validating the low-voltage avalanche behavior (relative to Si, GaAs), in overall, the study may provide some room for advancing our understanding in avalanche breakdown phenomena in 2D materials. Therefore, it would be worth exposing the study now to the entire field for discussion. In this respect, I am happy to recommend the article for publication.

Reviewer #2 (Remarks to the Author):

There are still some questions.

- 1) Authors have made a new WSe₂/hBN diodes, measured them, performed a high-resolution TEM, and claimed that the WSe₂/hBN diode showed better results than the original WSe₂. Why did he not add data to the manuscript and supplementary session? I don't understand why authors only use Fig.R1-R5 and Table R1 to answer my question, despite the opportunity to include better sample data in the Manuscript.
- 2) The fact that the overall performance of the WSe₂ diode clearly improves when using the hBN layer as the substrate proves that the reduction of the scattering process has a significant impact, as the authors claim, and as I and Reviewer 1 expected. So wouldn't a WSe₂ diode fully encapsulated with hBN have lower phonon scattering and thus better performance? Here is a good paper on disorder study in 2D heterostructure devices: Rhodes, Daniel, et al. "Disorder in van der Waals heterostructures of 2D materials." *Nature materials* 18.6 (2019): 541-549. Since the authors wrote an ambitious title, "the threshold limit of avalanche", I couldn't help but wonder how threshold could get better.
- 3) The authors still did not identify monolayer combinations.
- 4) The answer to my third question regarding the comparison of WSe₂ and InGaAs avalanche diodes is good. In the manuscript's Conclusion session, there should be a discussion like this and a further explanation of future applications. Currently, only its potential as photovoltaic applications is emphasized.

Response to Reviewers' comments

We appreciate the referees for their valuable comments on the manuscript. We have carefully addressed all the comments and made revisions accordingly. In this response letter, the referees' comments are presented in black italic font, while our responses are in regular blue font. All major changes have been highlighted in red in the main text and Supplementary Information.

Comments from the reviewers:

Reviewer #1 (Remarks to the Author):

In the revised manuscript, the authors have fully revised their manuscript and performed various extra measurements to support their claims on the near-threshold avalanche breakdown behavior in WSe₂ homojunction devices. Although I feel that the study lacks fundamental accounts for validating the low-voltage avalanche behavior (relative to Si, GaAs), in overall, the study may provide some room for advancing our understanding in avalanche breakdown phenomena in 2D materials. Therefore, it would be worth exposing the study now to the entire field for discussion. In this respect, I am happy to recommend the article for publication.

Response: Thanks for your positive comments. Two-dimensional transition metal dichalcogenide (TMD) material is more efficient than the bulk material in activating charge carrier avalanche, as evidenced by the theoretical-limit breakdown voltage, $V_{br}=1.6 V \approx E_g/q$. This characteristic allows avalanche devices to be operated just like a diode and thus can be integrated into the electronic-photonic heterogeneously-converging integrated circuits. This versatility opens up a wide range of potential applications.

In this work, we also show the fundamental image that gives rise to the low threshold avalanche phenomenon (Figure 1 of the main manuscript). Firstly, the reduced dimension of TMD materials results in fewer electron-phonon scatterings, minimizing energy loss during electron acceleration. Secondly, the device is shaped into a stepwise geometry that enhances the internal electric field 4 times. The combined effect of these factors leads to the low threshold avalanche behavior observed.

Reviewer #2 (Remarks to the Author):

There are still some questions.

1) Authors have made a new WSe₂/hBN diodes, measured them, performed a high-resolution TEM, and claimed that the WSe₂/hBN diode showed better results than the original WSe₂. Why did he not add data to the manuscript and supplementary session? I don't understand why authors only use Fig.R1-R5 and Table R1 to answer my question, despite the opportunity to include better sample data in the Manuscript.

Response: Thanks for your suggestions. We have incorporated the data on WSe₂/hBN devices, including the I - V curves, optical images, and TEM images, in the Supplementary Information (Supplementary Figures 21-25). Additionally, we have included a paragraph in the main manuscript that details the structure, fabrication procedure, and optoelectronic properties of the WSe₂/hBN device. Furthermore, we have conducted a comprehensive comparative analysis of the device performance between the WSe₂/hBN device and the bare WSe₂ device. The added content in the main manuscript is shown below.

“In the devices present above, the WSe₂ material is exposed to the SiO₂/Si substrate, which might suffer from the scattering from the substrate⁴². To clarify this kind of issue, we fabricated additional WSe₂/hexagonal boron nitride (hBN) devices. As shown in Supplementary Figure 21, an hBN film was first transferred onto the SiO₂/Si substrate, followed by a dry transfer of a stepwise WSe₂ layer. In this way, the WSe₂ material is isolated from the substrate. The electrode configuration is the same as that of bare WSe₂ devices. As summarized in Supplementary Figures 22-25, the breakdown voltage of all 11 WSe₂/hBN devices falls in the range of -1.2 to -1.8 V, while that of bare WSe₂ devices shows a quite discrete distribution in a wide voltage range, -1.4 to -5.4 V. For a more detailed analysis, we divided the threshold voltage into three ranges: $|V_{br}| < 1.5$ V, 1.5 V $\leq |V_{br}| < 2$ V, and $|V_{br}| \geq 2$ V. Among them, the proportions of WSe₂/hBN and pristine WSe₂ diodes in the three threshold voltage ranges are 36%, 64%, 0%, and 20%, 40%, 40%, respectively. This further confirms that the overall performance of WSe₂ diodes is indeed improved when an hBN layer is used as the substrate, due to the reduced scattering processes.”

2) The fact that the overall performance of the WSe₂ diode clearly improves when using the hBN layer as the substrate proves that the reduction of the scattering process has a significant impact, as the authors claim, and as I and Reviewer 1 expected. So wouldn't a WSe₂ diode fully encapsulated with hBN have lower phonon scattering and thus better performance? Here is a good paper on disorder study in 2D heterostructure devices: Rhodes, Daniel, et al. Since the authors wrote an ambitious title, "the threshold limit of avalanche", I couldn't help but wonder how threshold could get better.

Response: Thanks for your suggestion. In line with findings from the literature (Rhodes, D., Chae, S. H., Ribeiro-Palau, R., & Hone, J. Disorder in van der Waals heterostructures of 2D materials. *Nat. Mater.* 18, 541–549 (2019)), it is noted that SiO₂ is not an optimal substrate for van der Waals avalanche devices due to the inherent phonon scattering that can impede charge carrier transport and delay the avalanche process. Our experimental results confirm this aspect. As detailed in Supplementary Figures 22-25, the breakdown voltage of all 11 WSe₂/hBN devices falls within the range of -1.2 to -1.8 V, while the bare WSe₂ devices exhibit a more widely distributed breakdown voltage range, varying from -1.4 to -5.4 V.

We agree with your comment that there is still the possibility to enhance the performance of the WSe₂ avalanche diode, such as through complete encapsulation with an hBN layer. However, in principle, there may be limited room for improvement. This is because van der Waals materials lack surface dangling bonds, which means that surface passivation with a covering hBN layer may not result in a significant performance boost. Nonetheless, we plan to fabricate devices with this configuration and compare them with the current devices in future studies.

3) The authors still did not identify monolayer combinations.

Response: The concept of a “Monolayer” device holds great appeal in electronic chip design as it signifies scaling down each component to the physical limit, leading to an unprecedented level of integration. However, in the realm of photoelectronic chips, the “Monolayer” configuration is not as widely embraced. This is because the scaling-down process often comes at the expense of light absorption/emission efficiency, which can

significantly degrade the device's performance. For this reason, we do not prioritize the concept of a monolayer avalanche photodetector.

To prevent any potential confusion, we have explicitly mentioned the thickness of the WSe₂ avalanche devices in the revised manuscript. “We fabricated more than 25 devices based on different thickness combinations, in which the few-layer thickness spans from 3 L to 13 L and the multi-layer thickness spans from 13 L to 75 L. A statistical analysis of the geometrical configuration and photoelectric performances of those devices can be found in Supplementary Figures 3-10.”

4) The answer to my third question regarding the comparison of WSe₂ and InGaAs avalanche diodes is good. In the manuscript's Conclusion section, there should be a discussion like this and a further explanation of future applications. Currently, only its potential as photovoltaic applications is emphasized.

Response: Thanks for your suggestion. We have added a paragraph in the Discussion section to summarize the figure of merit of WSe₂ devices and compare them with that of the traditional counterpart. Additionally, we have elaborated on the potential opportunities they present for future optical communication.

“In summary, we propose and demonstrate a low-threshold stepwise WSe₂ avalanche diode structure. It is fabricated on a stepwise WSe₂ flake by depositing symmetric electrodes on the upstairs and downstairs layers. In such a device, the avalanche threshold voltage is lowered to ~1.6 V at room temperature and thus can be operated by an ordinary digital circuit (the driving voltage range is ± 5 V). The traditional counterpart, by contrast, cannot function without the high voltage accessories (the driving voltage is up to 150 V). Also, the WSe₂ device shows a low dark current (10-100 fA) in the linear region, which is 4 orders of magnitude lower than that of traditional avalanche devices. At the same time, the WSe₂ device shows the capability to detect ultra-weak light signals, e. g. 24 fW in illumination intensity, and 7.7×10^4 in photon count. Together, those characteristics indicate the great potential of van-der-Waals avalanche devices in future low-cost optical communication scenarios.”